



# Uptake of nitric acid, ammonia, and organics in orographic clouds: Mass spectrometric analyses of droplet residual and interstitial aerosol particles

Johannes Schneider[1], Stephan Mertes[2], Dominik van Pinxteren[2], Hartmut Herrmann[2], Stephan Borrmann[1,3]

[1]Particle Chemistry Department, Max Planck Institute for Chemistry, 55128 Mainz, Germany
[2]Leibniz Institute for Tropospheric Research, 04318 Leipzig, Germany
[3]Institute for Atmospheric Physics, Johannes Gutenberg University, 55128 Mainz, Germany

*Correspondence to:* J. Schneider (johannes.schneider@mpic.de)

**Abstract.** Concurrent in-situ analyses of interstitial aerosol and cloud droplet residues have been conducted at the Schmücke mountain site during the Hill Cap Cloud Thuringia campaign in central Germany in September and October 2010. Cloud droplets were sampled from warm clouds (temperatures between $-3$ and $+16°C$) by a counterflow virtual impactor and the submicron-sized residues were analyzed by a C-ToF-AMS, while the interstitial aerosol composition was measured by an HR-ToF-AMS. During cloud-free periods the submicron out-of-cloud aerosol was analyzed using both instruments, allowing for intercomparison between the two instruments. Further instrumentation included black carbon measurements and optical particle counters for the aerosol particles as well as optical sizing instrumentation for the cloud droplets. The results show that under cloud conditions on average 85% of the submicron aerosol mass partitioned into the cloud liquid phase. Scavenging efficiencies of nitrate, ammonium, sulfate, and organics ranged between 60 and 100%, with nitrate having in general the highest values. For black carbon, the scavenging efficiency was markedly lower (about 24%). The nitrate and ammonium mass fractions were found to be markedly enhanced in cloud residues, indicating uptake of gaseous nitric acid and ammonia into the aqueous phase. This effect was found to be temperature dependent: At lower temperatures the nitrate and ammonium mass fractions in the residues were higher. Also, the oxidation state of the organic matter in cloud residues was found to be temperature dependent: The O:C ratio was lower at higher temperatures. A possible explanation for this observation is a more effective uptake and/or higher concentrations of low-oxidized water soluble volatile organic compounds, possibly of biogenic origin, at higher temperatures. Organic nitrates were observed in cloud residuals as well as in the out-of-cloud aerosol, but no indication of a preferred partitioning of organic nitrates into the aqueous phase or into the gas phase was detected. Assuming the uptake of nitric acid and ammonia in cloud droplets to be reversible, it will lead to a redistribution of nitrate and ammonium among the aerosol particles, leading to more uniform, internally mixed particles after several cloud passages.

## 1 Introduction

The role of clouds and aerosol in the climate system is generally considered to be of great importance, but there is a consensus that our knowledge and understanding of the detailed processes of aerosol-cloud interaction in cloud formation and cloud evolution is still not sufficient (Lohmann and Feichter, 2005; Boucher et al., 2013;





Fuzzi et al., 2015). One aspect of cloud research is their formation which requires particles on which the supersaturated water vapor can condense. Depending on chemical and microphysical properties, aerosol particles are more or less well suited to act as cloud condensation nuclei (CCN). The ability of a particle to act as a CCN is generally described by the Köhler theory which is a superposition of Raoult and Kelvin effect (Köhler, 1936; McFiggans et al., 2006; Farmer et al., 2015). Under conditions of increasing relative humidity, aerosol particles take up water and the contained inorganic salts dissociate into their ionic components (deliquescence). In general, the number of ions that are formed by this process determines the ability of the particle to reach critical supersaturation and diameter and thereby become activated as a cloud condensation nucleus (Kreidenweis et al., 2005). Thus, the size of an aerosol particle is usually more important for cloud activation than its chemical composition (Dusek et al., 2006).

The other aspect of aerosol-cloud interaction is altering of the aerosol properties by cloud processing. Cloud droplets may scavenge gaseous substances that can dissolve in water, for example nitric acid, leading to an enhancement of nitrate in the cloud droplets (Levine and Schwartz, 1982; Strapp et al., 1988; Cape et al., 1997; Hayden et al., 2008). Also nitric oxides (NOx) may dissolve in the droplets where they are oxidized to nitrate (Strapp et al., 1988), but this pathway is regarded to be too slow under ambient conditions (Seinfeld and Pandis, 2006). Sulfate can be incorporated into cloud droplets by different pathways: Droplet formation on sulfate-containing CCN (nucleation scavenging), direct uptake of gaseous sulfuric acid ($H_2SO_4$), impaction scavenging of interstitial sulfate containing aerosol particles (that were too small to form the original CCN), or by in-cloud oxidation of sulfur dioxide (SO2) to H2SO4. The latter is the only sulfate production process and can occur via reaction of SO2 with O3 or H2O2 (Herrmann et al., 2015), but also by transition metal catalysis, via reaction with O2 (Calvert et al., 1985; Bradbury et al., 1999; Harris et al., 2013; Harris et al., 2014). Uptake of water soluble VOCs (volatile organic compounds) in cloud droplets (Laj et al., 1997; Herrmann et al., 2015; McNeill, 2015; van Pinxteren et al., 2015) as well as the formation of secondary organic aerosol in the aqueous phase (aqSOA, Ervens et al. (2011); Ervens (2015)) can lead to enhanced organic mass concentration in cloud droplets compared to ambient aerosol. This is supported by previous observations of cloud droplet residue composition: Drewnick et al. (2007) found in a mountaintop cloud study that organics and nitrate had the highest mass concentrations in cloud residues. Data reported by Sorooshian et al. (2010) showed that mainly organic acids and other oxygenated species were higher in cloud droplets as compared to out of cloud and interstitial aerosol, indicating uptake of oxidized VOCs or the formation of secondary, oxidized organic compounds in the cloud phase.

It is not quite understood whether the compounds formed by the processes listed above fully remain in the aerosol phase after cloud evaporation or if certain compounds (those with high volatility or low solubility) will be, at least partly, released back in to the gas phase (Cape et al., 1997; Sellegri et al., 2003). In any case, addition of soluble inorganic or organic compounds to the aerosol by cloud processing is expected to enhance the CCN properties of the processed aerosol, which has recently been confirmed by experimental data (Henning et al., 2014).

This paper focuses on the measurement of cloud residuals, interstitial aerosol and out-of-cloud aerosol during a hill-cap cloud study. We used a combination of counterflow virtual impactor (CVI) and aerosol mass spectrometer (Aerodyne AMS), similar to previous experiments by various research groups (Sorooshian et al.,



2006; Drewnick et al., 2007; Allan et al., 2008; Hayden et al., 2008; Gioda et al., 2009; Sorooshian et al., 2010). However, in contrast to those studies we deployed of two co-located AMS instruments, thereby allowing for the simultaneous measurement of interstitial and residual particle composition. Another approach was adopted by Hao et al. (2013) who also measured cloud residual composition using an AMS, but indirectly by subtracting the interstitial aerosol from the total aerosol (including cloud droplets and interstitial particles).

## 2 Measurements

### 2.1 Field site description and campaign overview

The measurements were conducted during the HCCT-2010 (Hill Cap Cloud Thuringia 2010) experiment between September 13 and October 25, 2010 (Tilgner et al., 2014). The experiment was carried out in the mountainous region "Thüringer Wald" (Thuringian Forest) in central Germany. Three field stations were set up for the experiment: an upwind station (Goldlauter, 905 m a.s.l.), a summit station (Schmücke, 938 m a.s.l.) and a downwind station (Gehlberg, 732 m a.s.l.). The sites were chosen based on the experiences of the experiment FEBUKO (Field Investigations of Budgets and Conversions of Particle Phase Organics in Tropospheric Cloud Processes) that was conducted at the same three sites in the years 2001 and 2002 (Herrmann et al., 2005). During the operation period of HCCT-2010, the summit station was covered in clouds (cloud liquid water content > 0.1 g m$^{-3}$) during 272 hours during the whole time period, corresponding to about 27% of the total measuring time (~ 1000 h). The prevailing wind direction measured locally at the summit site was SW (225 - 240°).

Here we focus on the simultaneous in-situ chemical analysis of the interstitial aerosol and the cloud residuals using two Aerodyne-type aerosol mass spectrometers at the summit site, and on the comparison of these data to out-of-cloud aerosol particles under comparable conditions. For the comparison of cloud residuals and interstitial aerosol, we have chosen the "Full Cloud Events" (FCE) defined in Tilgner et al. (2014) and listed in Table 1. For comparison of cloud residuals and out-of-cloud aerosol, we tried to find appropriate cloud-free comparison periods with similar air mass origin and close as possible in time to the cloud measurements. For this, we inspected the HYSPLIT backward trajectories (Stein et al., 2015; Rolph, 2016) that were calculated for the HCCT-2010 campaign on an hourly time scale (details given in the supplement to Tilgner et al. (2014)). For each FCE, a cloud-free period ("Non-Cloud Event", NCE) was chosen. The backward trajectories for all FCE and NCE are given in Figure 1. In some cases the NCE used here were identical to those defined in Tilgner et al. (2014), in other cases new events had to be defined. The exact times of the NCE are given in Table 1. The trajectories were inspected manually and rated from "-" to "+++", according to their similarity between the FCE and the NCE trajectories. The similarities of the trajectories will be taken into account when comparing the analysis of cloud residuals, interstitial aerosol and out-of-cloud aerosol.

### 2.2 Aerosol and cloud sampling at the summit site

The summit site used the laboratory in the top levels of a building owned by the German Environmental Protection Agency (Umweltbundesamt, UBA). The inlets were mounted into the windows of the uppermost level, directly below the roof at an altitude of approximately 15 m above ground, facing South-West (215°)





(Mertes et al., 2005b). The instruments for in-situ aerosol analysis were situated on the level below. Additional instrumentation for the cloud microphysics and cloud water sampling, as well as the meteorological station, were mounted on a 20 m high tower in about 20 m distance to the UBA building, and on the roof of the UBA building, respectively. Pictures of the sampling location can be found in the supplementary material (Figure S1).

Two inlets were used to sample aerosol and cloud droplets: An interstitial inlet with a cut-off aerodynamic diameter ($d_{aero}$) of 5 µm and a counterflow virtual impactor (CVI) in the same set-up and configuration as in the FEBUKO experiment (Mertes et al., 2005b). The CVI samples only cloud droplets larger than 5 µm and evaporates the cloud water using dry (RH < 10%) and particle-free carrier air. The remaining cloud droplet residues (CDR) can then be transferred to the various online analysis instruments. Due to the circumstances

described above the length of the sampling lines from the inlets to the instruments was in the order of several meters with vertical and horizontal sections. The sampling line losses were therefore calculated using the Particle Loss Calculator (von der Weiden et al., 2009) in an updated version including a pressure dependence to account for the 900 hPa ambient pressure at the site. The results yielded an aerodynamic diameter range with > 90% transmission for the interstitial sampling line of 20 nm – 950 nm (> 50% transmission between

4 nm and 2.5 µm), while for the CVI sampling line the > 90% transmission range was 10 nm – 1.8 µm (> 50% transmission from 2.5 nm to 3 µm). Both sampling lines had zero transmission for particles with aerodynamic diameter larger than 5 µm. This is no issue for the mass spectrometer measurements presented here, because the interstitial inlet itself has a cut-off of 5 µm, while the CVI samples the cloud droplets with $d_{aero}$ > 5 µm directly from the ambient air and only the residual particles are transferred to the instruments via the sampling lines.

**2.3 Analysis instruments**

We operated two Aerosol Mass Spectrometers (AMS) at the summit site: A C-ToF-AMS for cloud residual analysis and an HR-ToF-AMS for interstitial aerosol analysis. (For details see e.g., Drewnick et al., 2005; DeCarlo et al., 2006; Canagaratna et al., 2007). Additionally, a laser ablation aerosol mass spectrometer ALABAMA (Brands et al., 2011) was operated at the summit site sampling cloud residuals and out-of-cloud

aerosol (Roth et al., 2016). The black carbon content of the particles was determined using a Multi-Angle Absorption Photometer (MAAP, model 5012, Thermo Scientific) for the interstitial aerosol and two Particulate Soot Absorption Photometers (PSAP, Radiance Research), one for interstitial particles and the other for cloud residual particles. Particle size distributions of the interstitial aerosol particles and the cloud residuals were measured using scanning mobility particle sizers (SMPS, custom built at TROPOS) and optical particle counters

(OPC, model 1.109 and 1.108, Grimm Aerosol Technik, Germany). During cloud free times, the C-ToF-AMS and one OPC were switched manually to the interstitial inlet (now acting as an aerosol inlet) for instrument comparison (see section 2.4).

Cloud microphysical cloud parameters were determined outside the laboratory on a 20 m high tower. The cloud liquid water content (LWC) and effective radius were measured by a particle volume monitor (PVM, Gerber

(1991)), and cloud droplet number and size distribution were measured by an FSSP-100 (Dye and Baumgardner, 1984).





Meteorological weather parameters (wind, temperature, pressure, humidity, solar radiation) were recorded using a Davis Vantage Pro weather station (Davis Instruments, Hayward, CA) which was also mounted on the top of the tower. Temperatures ranged between minus 3 and 16°C, with the higher values at the beginning of the campaign and values below zero only occurring after October 12. Ambient pressure at the summit site ranged

between 890 and 915 hPa. Ambient relative humidity values reached 100% during cloud events, while the lowest values encountered were about 20% (October 09). The basic meteorological parameters that were measured by the Vantage Pro weather station are shown in Figure S2. A full overview of the meteorological conditions during HCCT2010 is given in the supplement to Tilgner et al. (2014).

### 2.4 Instrument calibrations and data quality assurance

The FSSP was calibrated before the field campaign with 15 µm borosilicate glass spheres. The droplet sizes were calculated using Mie theory for the refractive index of water (1.33), resulting in 14 size channels between 1 and 47 µm. The FSSP was actively pumped and the air flow speed through the instrument was determined using a hot wire anemometer to about 50 m/s.

The flow through the MAAP was set to 8 liters per minute to be consistent with the two other MAAP

instruments operated at the upwind and downwind station. The conversion from absorption to black carbon mass concentration was done for the MAAP with the manufacturer algorithm and for the PSAP using first the correction by Bond et al. (1999) and then applying a mass specific absorption cross section of 14.7 $m^2 g^{-1}$ according to Mertes et al. (2004). Since no scattering coefficient was measured, this part of the Bond correction had to be omitted, but PSAP filters were already changed at a transmission of 0.7 to minimize the scattering

artefact. Following Petzold et al. (2013), the values measured by MAAP and PSAP are reported here as "equivalent black carbon" (EBC).

The enrichment factor of the CVI is given by the ratio of the air flow in the CVI wind-tunnel to the sample flow inside the CVI inlet. Since both quantities are measured, the enrichment factor can be calculated. The sampling efficiency of the CVI is determined by comparing the number of residual particles counted behind the CVI and

the number of cloud droplets measured outside and by comparing the LWC measured in the CVI sampling line and the LWC measured outside. Both the enrichment factor and sampling efficiency were provided as a function of time and have been applied to the data presented here.

The aerosol mass spectrometers at the summit site were size calibrated with PSL particles, their ionization efficiency was calibrated with size selected ammonium nitrate particles, and the relative ionization efficiency

(RIE) for sulfate was determined using ammonium sulfate. These calibrations were done simultaneously with both instruments using the same test particles six times during the field campaign. The determined RIE values are given in Table S1 in the supplementary material. The collection efficiency (CE) was set to 0.5 for both mass spectrometers. The inlet flow was calibrated under ambient pressure conditions, such that all reported mass concentrations refer to ambient pressure (ranging between 890 and 915 hPa at the field site, see Figure S2 in the

supplement).



A prerequisite for the present analysis is the comparability of the instruments that were operated in parallel at the summit site, especially of the C-ToF-AMS and the HR-ToF-AMS. During cloud events, the C-ToF-AMS was used for the analysis of the cloud residuals and the HR-ToF-MS for interstitial aerosol. During non-cloud phases both instruments as well as the optical particle counters were connected to the interstitial inlet (now acting as an

aerosol inlet). A six-day cloud free period was chosen for the comparison between both mass spectrometers and between the two optical particle counters. Furthermore, the EBC concentrations measured by the two instruments that were always operated at the interstitial inlet (the MAAP and one PSAP) and the two optical particle counters were compared. Figure 2a) depicts the time series of the measured parameters during the cloud-free intercomparison period (19.09.2010 – 25.09.2010). The large variation of the atmospheric concentrations

(e.g., the organic aerosol mass concentration varies between < 1 and > 9 $\mu g\ m^{-3}$) confirms that this period is well suited for instrumental comparisons. Figure 2b) shows the correlation plots for the mass concentrations of EBC, of the main species measured by the AMS (sulfate, nitrate, organics, and ammonium), and of the particle number concentrations measured by the two OPC. The slopes, offsets and correlation coefficients are given in the graphs. In general the agreement between the two AMS both instruments is very good: For sulfate, organics, and nitrate

the slopes are between 0.977 and 0.992 with $r^2$ values between 0.945 and 0.985, only for ammonium there is a slight difference between the instruments, with a slope of 1.199 and $r^2$ of 0.966. Overall, this intercomparison confirms that comparisons between the interstitial and cloud residual particle composition is possible and differences that are larger than the differences during the intercomparison can be considered as significant. The two optical particle counters (shown is total number concentration for d > 250 nm) show excellent agreement

(slope = 0.990, $r^2$ = 0.9986. For EBC, the PSAP shows slightly smaller concentrations than the MAAP (slope = 0.838, $r^2$ = 0.951), but overall the agreement between both methods is satisfactory. Thus, we assume that also the second PSAP that was connected to the CVI throughout the campaign is in agreement with the MAAP. The full data sets (whole campaign measurement period) is shown in the supplementary material (Fig. S9).

In the AMS analysis, the signal intensity at m/z 44 ($CO_2^+$) and the ratio of the aerosol mass concentration

calculated from m/z 44 to the total organic aerosol mass concentration (typically denoted as $f_{44\ or}\ f_{CO_2^+}$) have been recognized as an indicator for oxygenated aerosol (Zhang et al., 2005; Ng et al., 2010; Sorooshian et al., 2010). Furthermore, $f_{44}$ has been used to infer the O:C ratio from high-resolution and unit mass resolution data (Aiken et al., 2008; Canagaratna et al., 2015). The contribution from gas-phase $CO_2$ to m/z 44 has to be corrected in the fragmentation table (Allan et al., 2004) during data evaluation. This contribution was determined using pure gas

phase measurements realized by adding a particle filter to the sampling. This correction has been applied to both instruments. For the C-ToF-AMS this had to be done separately for the interstitial inlet and the CVI sampling times, because the $CO_2$ content in the CVI sampling line is lower than in ambient air due to absorption of $CO_2$ in the molecular sieve that is used to remove $H_2O$ from the compressed air that is used to generate the dry carrier flow.

It has recently been demonstrated that different instruments do not agree in terms of the $f_{44}$ value (Fröhlich et al., 2015). This is supposed to originate from slight differences in the residence times of the molecules between vaporization and ionization, leading to a different extent of decarboxylation reactions. A similar observation was made during HCCT when comparing the $f_{44}$ values from C-ToF-AMS and HR-ToF-AMS during the





intercomparison period: The C-ToF-AMS showed systematically higher $f_{44}$ values (Figure S3 in the supplement) while the total organic mass concentration agreed (Figure 2). The parameterizations used to calculate O:C from $f_{44}$ (Aiken et al., 2008; Canagaratna et al., 2015) were derived from HR-ToF-AMS data. Thus, we chose to scale the $f_{44}$ from the C-ToF-AMS to the HR-ToF-AMS as shown in Figure S3, such that we can expect that the $f_{44}$

values are now comparable also for comparison of cloud residuals to interstitial aerosol, and that the O:C ratios inferred from $f_{44}$ are reliable. For the corresponding ratio $f_{43}$, the significance of which is discussed in Section 3.2.3., the necessity for such scaling was only marginal as the signals were near each other within the uncertainties (Figure S3).

## 3 Results and discussion

### 3.1 Cloud properties

The whole time series of the cloud droplet number distribution measured by the FSSP is given in Figure S4. The 14 full cloud events are also indicated by the grey bars. During FCE22.0 and FCE 24.0 the FSSP was not operational. The averaged number size distributions of the other 12 FCE are shown in Figure S5. During all full cloud events, the cloud droplet size distribution peaks between 10 and 20 μm in diameter. Smallest droplets are

around 3 μm, while larger droplets up to 40 μm are always present and the size range of the FSSP limits the detection of larger droplets. These values lie in a size range that has been reported from many previous hill cap cloud experiments (e.g., Wobrock et al., 1994; Cederfelt et al., 1997; Choularton et al., 1997; Hallberg et al., 1997; Martinsson et al., 1997; Wieprecht et al., 2005). Conversion of the number size distribution to total volume density and thereby to liquid cloud water content (LWC) yielded fair agreement to the PVM data

(slope = 0.80, $r^2$ = 0.66), but the conversion of a number size distribution to a total mass concentration is always subject to uncertainties. Therefore, the LWC reported by the PVM is regarded to be the more reliable quantity. The averaged values of the cloud parameters as LWC, number concentrations, droplet surface area, and droplet volume concentration for all full cloud events are summarized in Table 2. The averaged liquid water content (Tilgner et al., 2014) ranges between 0.14 and 0.37 g m⁻³, similar to values measured during the FEBUKO

experiments at the same site (Mertes et al., 2005a). The averaged cloud droplet number concentrations range between about 150 and 270 cm⁻³. These values, and also the LWC values, are on the low side of the ranges reported from previous hill cloud experiments (Martinsson et al., 1999; Bower et al., 2000; Mertes et al., 2005a).

### 3.2 Aerosol partitioning and composition during cloud events

As outlined above, the analysis presented here focuses on the full cloud events listed in Table 1. The mass

concentrations of the species organics, sulfate, nitrate, ammonium, chloride, and EBC measured during the full cloud events are given in Figure 3, separated for cloud residues (measured by the C-ToF-AMS and the PSAP using the CVI inlet) and for the interstitial aerosol (measured by the HR-ToF-AMS and the MAAP using the interstitial inlet). Additionally, the aerosol composition is given for the cloud-free comparison periods as explained above (see Table 1 and Figure 1), measured also with the HR-ToF-AMS and the MAAP. The data set

shows that in general the mass concentration of the interstitial aerosol is markedly lower than the cloud residue





mass concentration for all species except for EBC for which the interstitial mass concentration is generally higher than that of the cloud residuals. In most cases organic matter is the highest mass fraction, but the nitrate fraction is clearly enhanced in the cloud residues compared to interstitial and out-of-cloud data, such that in two cloud events (FCE22.2 and FCE24.0) the nitrate concentration exceeds the organic mass concentration in the
cloud residues. It is also interesting to note that in most cases the sum of interstitial aerosol and cloud residue concentration is higher that the out-of-cloud aerosol, especially for nitrate, but to a lesser degree also for ammonium and organics, indicating efficient uptake of these species by the cloud droplets. The only case where out-of-cloud nitrate (and also all other species) is larger than the sum of interstitial and residues is FCE26.1, for which the trajectories for the cloud event and the out-of-cloud comparison period are somewhat different (Figure
1). For sulfate, the picture is different: In several cases where nitrate is larger in-cloud than out-of-cloud, sulfate is lower (FCE5.1, FCE7.1, FCE11.2, FCE26.2). For organics, this is only the case in one event (FCE26.2). Thus the uptake of nitric acid, ammonia, and gaseous organic compounds by the cloud droplets appears to be more efficient than sulfate production from $SO_2$ oxidation, the scavenging of gaseous $H_2SO_4$ or the uptake of small sulfate particles.

**3.2.1 Uptake of nitric acid and ammonia**

Figure 4 shows the mass fractions of organics, sulfate, nitrate, ammonium, and EBC in the submicron aerosol (sum of all species detected by the AMS plus back carbon) in the residual and interstitial particles for each FCE, along with the mass fractions measured in the cloud-free comparison periods. For organics (upper panel) the mass fractions in interstitial, residues and out-of-cloud data are similar, with a slight trend towards lower mass
fractions in the residuals (with four exceptions). The sulfate mass fraction (second panel) in the residuals lies in most cases between the interstitial and out-of-cloud fraction, whereas the mass fraction in the out-of-cloud comparison periods is highest in nine out of 13 FCEs. In contrast, the mass fractions of nitrate (third panel) and also ammonium (forth panel) are higher in the cloud residuals than in the other data. EBC (lowermost panel) shows a complete different behavior: Here the mass fraction is highest in the interstitial aerosol and lowest in the
cloud residuals.

The higher nitrate and ammonium mass fractions are observed in almost all cloud events, with only two exceptions: FCE7.1, where the nitrate fraction is the same for residual and interstitial particles, and FCE13.3, where the aerosol in the cloud-free comparison period shows a higher nitrate fraction. Both events were rated with "++" with respect to the air mass origin. On average the nitrate fraction in the interstitial and in the out-of-
cloud aerosol is almost the same (about 14-17 %), while in the cloud residuals it is about 30 %. The corresponding values for the averaged ammonium fractions are 6% (interstitial), 10% (out-of-cloud), and 13% (cloud residuals). Together with the observation that also the absolute values of nitrate and ammonium are in most cases higher in the cloud residuals than in the out-of-cloud data (Figure 3), this finding suggests that the nitrate and ammonium enhancements in the cloud residuals are not an effect of different activation of ammonium
nitrate containing particles or of a higher ammonium nitrate fraction in larger particles. Instead it seems very likely that the additional nitrate and ammonium are present in the cloud droplets due to uptake of gaseous nitric acid ($HNO_3$) and ammonia ($NH_3$) into the liquid phase. Nitric acid is highly soluble (reported values for its





Henry's law solubility constant, $H^{cp}$, range between $8.8 \times 10^2$ and $2.6 \times 10^4$ mol m$^{-3}$ Pa$^{-1}$ (Sander, 2015)) such that this process is to be expected and has been identified and observed in numerous previous studies (Levine and Schwartz, 1982; Strapp et al., 1988; Cape et al., 1997; Sellegri et al., 2003; Tilgner et al., 2005; Drewnick et al., 2007; Hayden et al., 2008; Roth et al., 2016).

Ammonia has a Henry constant for solution in pure water of about $6.0 \times 10^{-1}$ mol m$^{-3}$ Pa$^{-1}$ (Sander, 2015) which much lower than that of nitric acid. However, acidic aerosol particles due to the solution of HNO$_3$ are likely to take up ammonia from the gas phase to neutralize the nitric acid and form NO$_3^-$ and NH$_4^+$ ions. We calculated the predicted ammonium in the aerosol particles assuming full neutralization of sulfate and nitrate and compared the values to the measured ammonium (Figure 5). There is no significant difference between the out-of-particles
and the cloud residuals, and it is found that in both cases the aerosol particles are fully neutralized. Thus we conclude that uptake of nitric acid with subsequent neutralization by ammonia is the reason for the enhanced nitrate and ammonium concentrations measured in the cloud residuals.

This interpretation contradicts the conclusions that we drew in a previous publication (Drewnick et al., 2007) for measurements of cloud residuals on the Swedish mountain Areskutan, where we argued based on measurements
before and after a cloud passage that we could exclude that the enhancement of nitrate or organics found in the residual particles is caused by scavenging of vapors by cloud droplets. However, in the light of the data presented here this conclusion may not have been valid. Hayden et al. (2008) who conducted aircraft-based measurements of cloud residuals speculated that most of the NO$_3^-$ entered the cloud water as HNO$_3$ and that the residual NO$_3^-$ measured may have been fixed by reaction with dissolved NH$_3$ or another buffer. Although the
authors do not report NH$_4^+$ concentrations in the cloud droplet, they concluded as well that the nitrate detected in the cloud droplets has most likely been in the form of ammonium nitrate.

Our data do not give information on the fate of the nitrate and ammonium after cloud evaporation. The situation in a CVI system is different from that in ambient air: The dry carrier air inside the CVI is dried by means of a molecular sieve that is designed to remove H$_2$O molecules but not HNO$_3$ and NH$_3$ molecules. Thus, the dry
carrier air is soon saturated with HNO$_3$ and NH$_3$, and thus the NO$_3^+$ and NH$_4^-$ will preferably remain in the particle phase, while in ambient air, dependent of the gas-phase concentrations of HNO$_3$ and NH$_3$, the situation will be different and a larger part of the nitrate and ammonium may be released back into the gas phase. If the air after the cloud returns to the same temperature and relative humidity conditions as before the cloud, is it to expect that the overall equilibrium between particle phase NH$_4$NO$_3$ and gas phase NH$_3$ and HNO$_3$ will be equal
to that before the cloud, as long as no chemical processing of nitrate and ammonium occurs in the cloud phase. Implications of this processes will be discussed in section 4.

The higher nitrate and ammonium fractions in the cloud events in the second half of the campaign when temperatures were lower indicate a temperature dependence. This is confirmed by Figure 6 that gives the mass fraction of nitrate in the CDR measured during the FCE as a function of temperature. A linear fit to the data
results in a significant correlation ($r^2 = 0.58$). Ammonium (not shown) is slightly less correlated with $r^2 = 0.32$. Although the Henry constant in general decreases with decreasing temperature (Sander, 2015) such that HNO$_3$ and NH$_3$ are dissolved better in the cloud water at higher temperatures, the thermodynamic equilibrium of the





nitric acid / ammonia / ammonium nitrate system is shifted towards dissociated nitrate and ammonium in an aqueous solutions (Seinfeld and Pandis, 2006) at lower temperatures. This holds as well for aerosol particles above deliquescence humidity, thus also the CCN will have a higher ammonium nitrate content at lower temperature. Both effects explain the observation of higher nitrate and ammonium fractions at lower

temperatures.

### 3.2.2 Scavenging efficiency

The comparison of the data measured during the cloud events (interstitial and residue mass concentration) with the out-of-cloud data that were measured at different times certainly bears the risk that different air masses with different aerosol properties and compositions are compared. Therefore, the following analyses will be based only

on the interstitial and cloud residue mass concentrations. From these two quantities, we calculated the scavenging efficiency $SE$, commonly defined as

$$SE = M_{\text{cloud residues}}/(M_{\text{cloud residues}} + M_{\text{intersitital}}) \qquad (1)$$

(Daum et al., 1984; Strapp et al., 1988; Kasper-Giebl et al., 2000; Hitzenberger et al., 2001), where $M$ is the measured mass concentration of the interstitial aerosol and of the cloud residues, respectively. The resulting $SE$

values for the full cloud events are shown in Figure 7. $SE$ is given for the total submicron aerosol mass as the sum of the non-refractive compounds plus EBC (a) and separated for the compounds nitrate, sulfate, organics, and EBC (b). Similar scavenging efficiencies are also used in the paper by van Pinxteren et al. (2016) in this special issue, not only as FCE averages but also time-resolved. Figure 7 shows that on average 85 % of the total submicron aerosol mass has partitioned into the cloud phase, with a maximum value of 94 % in FCE22.0 and

minimum values of about 66 % in FCE13.3. This partitioning can be the result of three different processes: activation of the pre-existing aerosol particles acting as CCN, scavenging of interstitial, non-activated aerosol particles, and uptake of gas-phase species by the cloud droplets as it was discussed for nitrate and ammonium above. Separation into the different species (Figure 7b) gives a more detailed picture: In general, nitrate and ammonium are the compounds with the highest $SE$ values while the organic compounds have the lowest $SE$, with

the exception of FCE7.1 where the SE of the organic aerosol is highest (91 %). In most cases the average $SE$ of sulfate lies between nitrate and organics, but during several cloud events (FCE 7.1, 11.2, 11.3, 13.3) sulfate has the lowest $SE$.

It has to be noted that $SE$ does not reflect the hygroscopicity of the pure compounds. Firstly, these values are averages over the whole measured range (AMS: approx. 40 – 700 nm, MAAP/PSAP: whole particle size range

transmitted through the sampling line) and over a large number of aerosol particles. Secondly, as shown by single particle measurements conducted during the same experiment by Roth et al. (2016), most of the aerosol particles were found to be internal mixtures, with especially sulfate and nitrate being present in almost all analyzed particles. Together with the fact that the amount of available ions in an deliquesced aerosol particle determines the CCN activity (e.g., Köhler, 1936; Kreidenweis et al., 2005) leading to a better activation of larger

particles (Dusek et al., 2006), it becomes clear that all particles having a sufficient amount of soluble material like nitrate or sulfate become activated. Thus, even if the organic content of these particles would be completely





hydrophobic, organics would be present in the cloud residuals. On the contrary, it can be seen from our data the organic compounds are mainly oxygenated and thereby moderately hydrophilic (Jimenez et al., 2009), thereby additionally increasing the activation efficiency of the particles.

A similar observation can be made for EBC. Our data show that the mass based *SE* of EBC is rather low with an
average value of about 24 %, although occasionally (FCE 7.1) an *SE* of 50 % was observed. The single particle analysis presented by Roth et al. (2016) shows an increased number fraction of soot-containing particles in the cloud residues. Interestingly, their soot-containing single particle number fraction inside the cloud residues is highest during FCE7.1, where also the mass based *SE* of EBC presented here shows the highest value with 50 %. The size dependent data by Roth et al. (2016) show that soot-containing particles are mainly observed at
diameters above 500 nm. Such large aged soot-containing particles, which are internally mixed with sulfate and nitrate, also very likely are activated as CCN. In contrast, small and fresh soot particles are usually smaller than 150 nm and are therefore not detected by the single particle instrument used by Roth et al. (2016). These particles will not be activated and remain in the interstitial aerosol (as can be seen from the higher mass fraction of EBC in the interstitial aerosol, see Figure 4), leading to the overall lower *SE* for EBC. The soot-containing
particles around 500 nm therefore presumably contain only small amounts of soot but are mainly composed of nitrate and sulfate.

We did not observe significant correlations of the scavenging efficiencies averaged for the individual FCE with the cloud parameters cloud liquid water content (LWC) and cloud droplet number concentration (CDNC). The data are shown in Figure S7 of the supplementary material. An explanation might be that the variation from one
cloud event to the other is too high to observe the effects of uptake from the gas-phase and in-cloud production. A time resolved analysis of one cloud event (FCE13.3) will therefore be presented in a later section (3.3).

### 3.2.3 Oxidation properties of the organic compounds in CDR

From Figure 7 it can be seen that the *SE* of the organics is almost as high as that of sulfate. This may be due to activation of particles containing organics due to internal mixture with nitrate and sulfate, but also due to soluble
organic material contained in the cloud forming particles. Furthermore, also uptake of water soluble VOCs from the gas phase into the cloud droplets may play a role. Such an uptake may either lead to a similar behavior as the uptake of nitrate and ammonium, namely a temporary shift of the equilibrium between gas phase and aqueous phase, such that after cloud evaporation the original equilibrium will re-establish. Another possibility is formation of secondary organic aerosol (SOA) in the aqueous phase, leading to the so-called aqSOA. aqSOA
would remain in the particle phase and thereby lead to an increase of the aerosol mass after cloud evaporation, similar to sulfate production from in-cloud oxidation of $SO_2$. The formation of aqSOA was first suggested by Blando and Turpin (2000) and later verified by a number of laboratory and field experiments (e.g., El-Sayed et al. (2015) and review by Ervens et al. (2011)). Observations have shown that organic acids occur in cloud residuals and it is thought that the conversion of water-soluble precursor species as glyoxal (ethanedial) to
organic acids is facilitated in the aqueous phase (Blando and Turpin, 2000; Sorooshian et al., 2010). Oxygenated organic compounds like organic acids have been found to be correlated with m/z 44 ($CO_2^+$) (Zhang et al., 2005; Ng et al., 2010; Sorooshian et al., 2010) and thus we will use the measured fraction of the organic signals at m/z





43 and m/z 44 ($f_{43}$ and $f_{44}$, see section 2.4) as an indicator for oxygenated organic compounds in cloud droplet residues in the following. It was previously shown that $f_{43}$ and $f_{44}$ can be used to represent of the oxidation level of organic aerosol (Ng et al., 2010; Ng et al., 2011). $f_{44}$ increases with the oxidation level of the organic aerosol, while $f_{43}$ decreases due to oxidation of $C_3H_7^+$ and/or $C_2H_3O^+$ ions (Lambe et al., 2011). Figure 8 shows $f_{44}$ as a

function of $f_{43}$ for the organic aerosol data measured in HCCT2010. The left panel shows the whole data set (15-min averages) including all out-of-cloud data, all cloud residues, and all interstitial data. For separation between out-of-cloud and interstitial data we used a lower threshold of the 15-min average value of the LWC of 0.1 g m$^{-3}$ for cloud conditions (aerosol particles are interstitial) and an upper threshold of 0.01 g m$^{-3}$ for non-cloud conditions (aerosol particles are out-of-cloud).

The bars represent the 25% and 75% percentiles to the median values, while the dotted lines indicate the range of atmospheric observations as reported by Ng et al. (2010). Out-of-cloud data have been measured with both instruments, although more data points were recorded with the HR-ToF-AMS than with the C-ToF-AMS. This was so because the C-ToF-AMS was connected to the CVI when a cloud was expected or when it had just disappeared, while the HR-ToF-AMS measured out-of-cloud data during all times when the LWC was below the

threshold. The $f_{43}$ and $f_{44}$ values of the C-ToF-AMS were scaled to the HR-ToF-AMS during the intercomparison period as explained in Section 2.4, such that it is not surprising that the out-of-cloud data (yellow and green) agree well between both instruments. The interstitial data reveal a lower $f_{44}$ and a higher $f_{43}$, thereby indicating a lower oxidation state as the out-of-cloud aerosol. Less oxidized organic compounds are in general less hygroscopic (Jimenez et al., 2009; Lambe et al., 2011), thus this it is not surprising that particles containing more

low-hygroscopic compounds are less likely activated in the cloud. In contrast, the cloud residue data reveal an unexpected behavior: The spread of the data in the $f_{44}$ space is much larger, and the data extend to very low values of both $f_{43}$ and $f_{44}$, partly lower than the atmospheric range reported previously. To closer investigate the large spread in the $f_{44}$ space, the right panel of Figure 8 shows only the CDR data, but color coded with ambient temperature. Also given are the mean values and standard deviations of the individual FCEs (black markers and

bars). The data are grouped in two regimes, and the color code shows that these two regimes are separated by temperature. The data with the lowest temperature (blue colors) lie in an area with higher $f_{44}$ values between 0.15 and 0.20, while the data measured at higher temperatures (green and yellow colors) fall in an $f_{44}$ regime between 0.01 and 0.12. Also in the $f_{43}$ space we observe a temperature dependence: $f_{43}$ increases with temperature. For a more detailed inspection of the temperature dependence and to exclude instrumental issues as a reason for this

finding, Figure 9 shows the $f_{44}$ and $f_{43}$ values as a function of temperature. Two FCE (24.0 and 22.0) occurred even at temperatures below zero, thus during these events supercooled clouds were probed. In both graphs the averaged values for the FCE are plotted along with all out-of-cloud data (15-min averages) measured with the C-ToF-AMS. On the right axis of the upper graph the approximate O:C ratio calculated via the "Aiken-Ambient method" (Aiken et al., 2008; Canagaratna et al., 2015) from the unit mass resolution data measured by the C-

ToF-AMS is indicated. From the upper panel it can be concluded that the low $f_{44}$ values in the cloud residuals at the higher temperatures (> 5°C) are not an instrumental artefact, because the out-of-cloud values measured at the same temperatures with the same instrument are much higher than the CDR data. The out-of-cloud values of $f_{44}$ for T > 5°C range between 0.10 and 0.22 (corresponding to O:C ratios between 0.5 and 1.0) while the $f_{44}$ values





in the CDR are between 0.04 and 0.1 (O:C between 0.2 and 0.6). At low temperatures (< 4°C), the organic aerosol in the out-of-cloud aerosol and the CDR have approximately the same $f_{44}$ values (0.15 – 0.20), corresponding to a O:C ratio of about 0.7 to 0.9. The relatively high O:C ratios of 0.7 to 0.9 indicate low volatile oxygenated organic aerosol (LV-OOA) while O:C ratios between 0.2 and 0.6 indicate semivolatile (SV-) OOA

(Crippa et al., 2013; Canagaratna et al., 2015).

In the $f_{43}$ data (lower panel) the temperature dependence is much less pronounced but visible at the highest temperatures (FCE1.1, FCE1.2, FCE13.3). Here, the out-of-cloud data correspond well to the CDR data.

Exemplary mass spectra for one FCE with high (FCE5.1) and one with low temperature (FCE24.0) are shown in Figure 10, along with the $f_{44}$ values, for interstitial aerosol, out-of-cloud aerosol, and cloud residuals. The two

events FCE5.1 and FCE24.0 were chosen because for both events the trajectories of the cloud-free comparison period matched very well with those from the cloud events (Figure 1 and Table 1). As noted above, the $f_{44}$ values measured in the interstitial aerosol are lower than those measured during the cloud-free comparison period, but the $f_{44}$ values of the residuals are markedly lower at higher temperatures (FCE5.1). The mass spectra show that the temperature difference causes different peak heights, but does not lead to additional organic ions. Besides

m/z 43, also other organic signals like m/z 29, m/z 41, and m/z 55 are higher in the mass spectrum recorded at higher temperatures. m/z 60, which is a typical marker for biomass burning (Schneider et al., 2006; Alfarra et al., 2007) is low in all mass spectra, but slightly more pronounced in the mass spectra of the interstitial and out-of-cloud aerosol particles at lower temperatures (FCE24.0 and NCE0.9).

A possible explanation for the observation that the $f_{44}$ values (and thus the O:C ratios) of the cloud residual

particles are so low at the higher temperatures (T > 5°C) is more efficient uptake of less oxidized organic compounds (low $f_{44}$) by the cloud droplets at higher temperatures, because solubility (Henry's law constant) generally increases with temperature (Sander, 2015). Also, it is conceivable that at higher temperatures such low-oxidized gas-phase compounds, as for example biogenic volatile organic compounds, are more abundant.

There is only a limited number of studies that investigated cloud residues using an AMS, three of which

(Drewnick et al., 2007; Hayden et al., 2008; Sorooshian et al., 2010) used a CVI to sample the cloud residual, while Hao et al. (2013) inferred the residue composition indirectly by subtracting total and interstitial aerosol. Hayden et al. (2008) did not investigate organics. In the study of Drewnick et al. (2007) that took place on a Swedish mountain in summer, no difference in $f_{44}$ between out-of-cloud aerosol and CDR was observed. In contrast, Sorooshian et al. (2010) report from aircraft studies that $f_{44}$ was enhanced in CDR. Also, Hao et al.

(2013) found a slight increase in LV-OOA (corresponding to higher $f_{44}$ values) in CDR. However, none of these studies investigated the influence of ambient temperature. Temperature can influence the cloud droplet composition not only through the solubility of VOCs but also through different emissions of VOCs as a function of temperature, e.g. higher biogenic emissions at higher temperatures and higher anthropogenic emissions (domestic heating) at lower temperatures.



### 3.2.4 Organic nitrates

Drewnick et al. (2007) and Hao et al. (2013) both analyzed the CDR for their nitrate ion ratio ($NO^+/NO_2^+$) which gives an indication for the presence of organic nitrate. Organic nitrates have been found to have a ratio $NO^+/NO_2^+$ (m/z 30 / m/z 46) between 5 and 15 (Fry et al., 2009; Bruns et al., 2010), while ammonium nitrate has

lower values with published values between 2 and 3 (Alfarra et al., 2006; Drewnick et al., 2007; Bruns et al., 2010; Hao et al., 2013). Although in our study the aerosol was fully neutralized within the uncertainties (Figure 5) which suggests that nitrate is always present in the form of ammonium nitrate, we will explore the $NO^+/NO_2^+$ ratio to assess the possibility of organic nitrate formation. The $NO^+/NO_2^+$ ratios measured using the ammonium nitrate calibrations were 3.1 for the HR-ToF-AMS and 3.3 for the C-ToF-AMS. In both cloud studies mentioned

above (Drewnick et al. (2007) and Hao et al. (2013)), the cloud residuals contained a lower amount of organic nitrates than the out-of-cloud aerosol. This was only partly the case in our study. The $NO^+/NO_2^+$ ratio of the CDR is shown Figure 11 along with the $NO^+/NO_2^+$ ratio of the out-of-cloud aerosol and the $NH_4NO_3$ calibration values. All data were measured with the C-ToF-AMS and therefore represent unit mass resolution data. $NO_2^+$ corresponds to the signal at m/z 46 while $NO^+$ was taken as m/z 30 corrected for the gas-phase ion signals and

organic ion fragments using the unit mass resolution fragmentation table (Allan et al., 2004). We observe a slight temperature dependence of the $NO^+/NO_2^+$ ratio of the out of-cloud aerosol: the values increase with temperature and reach the pure $NH_4NO_3$ value only at the lowest temperature (below 0°C). This finding suggests that the abundance of organic nitrate in the particle phase increases with temperature, which is in contradiction to recent observations by Lee et al. (2014) who showed that formation of organic nitrate is enhanced at lower

temperatures. Since also the equilibrium between particle phase and gas phase should be shifted towards the gas phase at higher temperatures, a higher amount of organic nitrates in the aerosol phase at higher temperatures does not agree with the observations by Lee et al. (2014).

In contrast to the aerosol data, our CDR data do not such a clear trend: part of the values match the out-of-cloud data, while a few values remain at the low ratio as measured for $NH_4NO_3$. Thus, only these five data points

confirm the finding of Drewnick et al. (2007) and Hao et al. (2013) that CDR contain a lower amount of organic nitrates than the out-of-cloud aerosol, while the other seven data points suggest no significant different organic nitrate content in CDR compared to the aerosol. One important pathway for organic nitrate formation is oxidation of organic precursors by reaction with the $NO_3$ radical (Fry et al., 2013) which is only present at nighttime. Thus we separated the FCE into night, day-and-night, and daytime events according to the FCE times

given in Table 1. Black data points in Figure 11 indicate night-time data, dark grey day-and-night data, and light grey daytime data (only FCE26.2 fell into the latter category). Some data points seem to support the hypothesis that night time clouds contain a larger amount of organic nitrates, but a few data points do not fit into this picture. Especially the only daytime cloud measurement (FCE26.2) shows a clearly elevated $NO^+/NO_2^+$ ratio compared to pure ammonium nitrate and to other FCE that were measured during nighttime. Thus, we conclude

that the variation of the organic nitrate amount in CDR probably is more dependent on the air mass origin, chemical composition of the particles, and the availability of organic precursor gases that can react to organic nitrates.



### 3.3 Case study: Onset and temporal evolution of a cloud

In the last section of this paper we examine the temporal evolution of the chemical and microphysical properties of the cloud droplets and their residuals. As an example we chose the cloud period on 05 – 07 October 2010, a period that includes FCE11.3. Figure 12 shows the cloud droplet size distribution (CDSD, measured using the

FSSP, upper panel), the cloud droplet number concentration (CDNC) along with the nitrate mass concentration of the CDR ($2^{nd}$ panel), the liquid water content (LWC, measured using the PVM) along with the organic mass concentration of the CDR ($3^{rd}$ panel), and the size distributions of nitrate and organics in the cloud residues ($4^{th}$ and $5^{th}$ panel) along with the modal diameter (from a unimodal lognormal fit). All data represent 10-min averages except for the CDSD which is given on a 5-sec time base. The data show that the cloud period can be

divided in two parts: Before and after 06 October, 12:00. The cloud droplet size increases from about 5 to 15 μm during the first part until about 12:00 (upper panel), then starts again at a smaller diameter (about 10 μm and increases up to 20 μm during the second half of the cloud until the cloud disappears at 03:35 on 07 October. During the first half, the CNDC ($2^{nd}$ panel) decreases, while it increases during the second half. Interestingly, the nitrate mass concentration of the CDR follows the CDNC only during the first half, with a correlation coefficient

of $r^2 = 0.81$ (Figure S8 in the supplement). In the second half, the correlation of CDR nitrate with CNDC is much weaker ($r^2 = 0.12$). The LWC (third panel) increases during both cloud parts although the CDNC decreases in the first half and remains approximately constant during the second half, because the droplets become larger in both parts of the cloud period. The organic mass concentration in the CDR follows the LWC slightly better in both parts (first half: $r^2 = 0.22$, second half: $r^2 = 0.35$, see Figure S8) than it follows the CDNC. The nitrate mass

concentration in the CDR shows no correlation at all with the LWC. The organic mass concentration in the CDR shows the highest concentrations later than nitrate ($4^{th}$ and $5^{th}$ panel).

As discussed above, there are two mechanisms that are responsible for nitrate in CDR: activation of nitrate-containing particles and uptake of nitric acid from the gas-phase by the cloud droplets. The first mechanism would certainly lead to a correlation between CDNC and CDR nitrate, assuming similar sizes and nitrate content

of the original CCN. For the second mechanism, this depends on whether the uptake of nitric acid from the gas phase is limited by its solubility or by the amount of nitric acid available in the gas phase. If it would be limited by solubility, then larger drops should take up more nitric acid and a correlation with LWC would be expected. This is apparently not the case. Thus we conclude that nitric acid uptake is not limited by its solubility. In the second phase of the cloud, where no correlation between CDR nitrate and CDNC is observed, this indicates that

not enough nitric acid was available in the air, such that even a growing number of cloud droplets and an increasing LWC could not lead to more nitrate in the droplets, because the gas phase is already depleted. But this is only a speculation, because no parallel gas-phase nitric acid measurements have been conducted at the measurement site.

The organic mass concentration in the CDR has slightly better correlations with LWC than with CDNC in both

phases. Following the reasoning above this may mean that uptake of organic species from the gas phase is more likely limited by the solubility of the compounds, such that larger droplets can take up larger amounts of organic compounds, leading to a correlation between CDR organics and LWC. FCE11.3 was one of the events at higher





temperature (Figure 9) where we observed the low $f_{44}$ values and had concluded that uptake of less oxidized organic compounds from the gas phase may occur. This hypothesis is now strengthened, although certainly not proven, by the observations during the cloud evolution in Figure 12.

Most of the time the modal diameter of the organic mass distribution equals that of the nitrate mass distribution
(around 600 nm). Only during the end of the first cloud part (between 6:00 and 12:00 on 06 October 2015), at the time when the LWC is highest, the modal diameter of the organic mass distribution decreases to about 500 nm, while the nitrate modal diameter remains unchanged. A vacuum aerodynamic diameter of 600 nm corresponds roughly to a volume equivalent diameter of 400 nm (assuming spherical particles and an average density of 1.5 g cm$^{-3}$), and a mass size distribution peak at 400 nm corresponds to a number size distribution
peak at about 320 nm (calculated via the Hatch-Choate equations (Hinds, 1999) using the distribution width from the lognormal fit of 1.32). The smaller modal diameter of the organic residues together with the higher LWC during this time may indicate that due to a higher supersaturation during this part of the cloud, smaller particles with a higher organic content have been activated. Typically critical activation diameters were in a range between 100 and 200 nm (Henning et al., 2014) during the HCCT2010 campaign, such that also this
finding agrees with the assumption of uptake from the gas-phase during cloud processing, leading to larger residue sizes than the original CCN sizes.

### 4 Summary and Conclusions

We have investigated the physico-chemical composition of cloud droplets and cloud droplet residues along with the composition of interstitial and out-of-cloud aerosol particles during a six-week field study at the German
mountain range "Thüringer Wald". During the cloud events most of the submicron aerosol mass (average 85%) has partitioned into the cloud phase and only 15% remained in the interstitial phase. The results give clear evidence for the uptake of nitric acid and ammonia in the CDR. The mass fraction of nitrate in the CDR was 30% on average, while in interstitial aerosol and during the out-of-cloud comparison periods it was between 13% and 17%. The finding that no significant depletion of nitrate is found in the interstitial aerosol leads to the
conclusion that nitrate addition to the cloud droplets occurs via uptake of nitric acid from the gas-phase, as has been observed previously in numerous studies (Levine and Schwartz, 1982; Strapp et al., 1988; Cape et al., 1997; Sellegri et al., 2003; Tilgner et al., 2005; Drewnick et al., 2007; Hayden et al., 2008; Hao et al., 2013), with subsequent neutralization by ammonia. The nitrate and ammonium fractions in the CDR showed a temperature dependence (higher content at lower temperatures) with a significant linear correlation of $r^2 = 0.58$
for nitrate and $r^2 = 0.32$ for ammonium. Scavenging efficiencies averaged over for all defined full cloud events (FCE) showed no clear correlation neither with liquid water content (LWC) nor with cloud droplet number concentration (CDNC). In a time resolved case study of a cloud event, nitrate was better correlated CNDC than with LWC, indicating that nitric acid uptake is limited by the availability of nitric acid and not by its solubility. Sulfate production by $H_2O_2$ and trace-metal catalyzed oxidation of $SO_2$ in the liquid phase was observed in the
same field experiment (Harris et al., 2013; Harris et al., 2014), but could not be detected with the approach presented here,, most likely because the nitrate enhancement is much stronger.



The scavenging efficiency of organics was lower than that of nitrate and ammonium, resulting in higher organic mass fractions in the interstitial aerosol than in the CDR in most FCE. Nevertheless, on average about 82% of the organic aerosol mass has partitioned into the cloud phase, and clear indications for uptake of organic compounds from the gas phase were found: We observed a temperature dependence of the oxidation properties
(O:C ratio) of the organic compounds in the CDR and conclude that at higher temperatures, uptake of low-oxidized compounds (lower $f_{44}$, higher $f_{43}$) is occurring, facilitated by higher solubility at higher temperatures, but possibly also due to higher abundance of low-oxidized organic compounds (e.g. from biogenic processes) in ambient air at higher temperatures.

Previous studies (Drewnick et al., 2007; Hao et al., 2013) observed that organic nitrates are found preferably in
the out-of-cloud aerosol but not in CDR. This was not confirmed by our study. We have observed neither a clear trend of the presence of organic nitrate as a function of out-of-cloud aerosol/CDR, nor a discernible temperature dependence and have concluded that air mass origin seems to determine the amount of organic nitrates.

The time resolved case study of cloud evolution shows a moderate correlation between the organic CRD mass concentration and the LWC, which also indicates uptake of organic compounds from the gas phase by the
droplets, but in contrast to nitrate, this uptake appears to be limited by the solubility, such that a higher absolute amount of available water is able to take up more water-soluble organics. For nitrate, as mentioned above, this appears not to be the case, because nitric acid is so highly soluble that already at fairly low liquid water contents $(0.1 - 0.2 \text{ g m}^{-3})$ the available nitric acid tends to be depleted from the gas phase, and the amount of nitrate mass concentration measured in the CDR is then proportional to the number concentration of droplets in the cloud.

In general, cloud processing will tend to evenly distribute nitrate and ammonium over the processed aerosol particles: At the same temperature and relative humidity after the cloud passage as before, it is to be assumed that the same equilibrium between particle-phase ammonium nitrate and gas phase nitric acid and ammonia as before the cloud is established. Thus, the absolute amount of particle phase ammonium nitrate should be the same after the cloud as before the cloud. But as our data have shown, all cloud droplets take up nitric acid and
ammonia, such that after cloud evaporation all released aerosol particles contain ammonium nitrate. After several cloud processes it is to expect that the available ammonium nitrate at a certain temperature and relative humidity is evenly distributed over all aerosol particles. For the water soluble organic compounds taken up by the cloud droplets the effect is expected to be similar, but here also chemical processing might occur leading to enhanced organic aerosol mass after cloud passage (similar to sulfate production by in-cloud oxidation of $SO_2$).
The redistribution of ammonium nitrate over all aerosol particles may explain the increase of the hygroscopicity of the aerosol particles that are released after the cloud has evaporated, as reported by Henning et al. (2014) from the same experiment.

**Supplement**

Supporting online material to this manuscript is available at:





**Author contribution**

H. Herrmann and D. van Pinxteren designed the experiment, J. Schneider, S. Mertes, and D. van Pinxteren carried out research, J. Schneider and S. Mertes evaluated the data. J. Schneider wrote the manuscript, with contributions by S. Borrmann, S. Mertes, and D. van Pinxteren.

**Acknowledgements**

We thank S. Günnel, P. Glomb, L. Schenk, W. Schneider, T. Böttger, and A. Roth for general support, W. Frey, S. Molleker, C. von Glahn, and T. Klimach for help with the FSSP operation and data acquisition, and F. Freutel, S.-L. von der Weiden-Reinmüller, J. Diesch, and P. Reitz for temporary instrument operation during the campaign. Additionally, we thank the German Federal Environmental Agency (UBA) for providing laboratory

space and field site, and the whole HCCT Team for support, data exchange and discussion. This work was supported by the Max Planck Society and the DFG, grants HE 3086/15-1 and HE 939/25-1.

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



**Table 1: Cloud events as defined in Tilgner et al. (2014), and adjacent time periods of non-cloud conditions, defined in the present study. FCE = Full Cloud Event, NCE = Non-Cloud Event. All times are given in local time.**

| Full cloud event time | | | Non-cloud comparison time | | | |
|---|---|---|---|---|---|---|
| Event | Start Time | End Time | Event | Start Time | End Time | Trajectory similarity |
| FCE1.1 | 14.09.10 11:00 | 15.09.10 01:50 | NCE0.1 | 15.09.10 12:10 | 15.09.10 22:10 | ++ |
| FCE1.2 | 15.09.10 03:00 | 15.09.10 06:20 | NCE0.1 | 15.09.10 12:10 | 15.09.10 22:10 | ++ |
| FCE2.1 | 15.09.10 23:00 | 16.09.10 02:00 | NCE0.1 & NCE0.2 | 15.09.10 12:10 16.09.10 02:50 | 15.09.10 22:10 16.09.10 06:10 | + |
| FCE4.1 | 16.09.10 13:10 | 16.09.10 15:00 | --- | --- | --- | |
| FCE5.1 | 16.09.10 21:40 | 16.09.10 23:50 | NCE5.1 | 16.09.10 16:40 | 16.09.10 18:20 | +++ |
| FCE7.1 | 24.09.10 21:10 | 25.09.10 00:50 | NCE0.7 | 24.09.10 10:30 | 24.09.10 20:00 | ++ |
| FCE11.2 | 01.10.10 20:50 | 02.10.10 03:10 | NCE0.8 | 03.10.10 11:20 | 03.10.10 14:40 | + |
| FCE11.3 | 02.10.10 07:10 | 03.10.10 00:30 | NCE0.8 | 03.10.10 11:20 | 03.10.10 14:40 | + |
| FCE13.3 | 06.10.10 06:50 | 07.10.10 01:00 | NCE13.3 | 07.10.10 04:00 | 07.10.10 07:00 | ++ |
| FCE22.0 | 19.10.10 01:50 | 19.10.10 09:00 | NCE22.0 | 18.10.10 15:00 | 18.10.10 21:00 | + |
| FCE22.1 | 19.10.10 21:10 | 20.10.10 02:30 | NCE22.1 | 20.10.10 14:30 | 20.10.10 18:00 | - |
| FCE24.0 | 21.10.10 22:10 | 22.10.10 10:00 | NCE0.9 | 21.10.10 12:40 | 21.10.10 21:20 | +++ |
| FCE26.1 | 23.10.10 23:40 | 24.10.10 07:20 | NCE0.10 | 23.10.10 16:10 | 23.10.10 22:40 | + |
| FCE26.2 | 24.10.10 08:40 | 24.10.10 12:20 | NCE26.2 | 24.10.10 13:00 | 24.10.10 18:00 | +++ |



**Table 2: Averaged meteorological parameters and cloud properties measured during the full cloud events, along with standard deviations. LWC: Liquid water content, NCD: Cloud droplet number concentration, SCD: Cloud droplet surface concentration, VCD: Cloud droplet volume concentration.**

| | Cloud Event | Temperature | Pressure | LWC (PVM) | $N_{CD}$ (FSSP) | $S_{CD}$ (FSSP) | $V_{CD}$ (FSSP) |
|---|---|---|---|---|---|---|---|
| | | °C | hPa | g m⁻³ | cm⁻³ | μm² cm⁻³ | cm³ m⁻³ |
| 1 | FCE1.1 | $9.2 \pm 0.9$ | 907 | $0.25 \pm 0.12$ | $272 \pm 31$ | $8.2 (\pm 2.4)e+04$ | $0.16 \pm 0.09$ |
| 2 | FCE1.2 | $9.2 \pm 0.1$ | 901 | $0.20 \pm 0.07$ | $260 \pm 21$ | $7.0 (\pm 2.5)e+04$ | $0.13 \pm 0.14$ |
| 3 | FCE2.1 | $6.8 \pm 0.1$ | 898 | $0.17 \pm 0.07$ | $203 \pm 29$ | $5.7 (\pm 1.8)e+04$ | $0.10 \pm 0.77$ |
| 4 | FCE4.1 | $6.9 \pm 0.1$ | 900 | $0.13 \pm 0.05$ | $156 \pm 21$ | $4.8 (\pm 1.2)e+04$ | $0.09 \pm 0.30$ |
| 5 | FCE5.1 | $6.9 \pm 0.1$ | 900 | $0.30 \pm 0.04$ | $188 \pm 14$ | $9.4 (\pm 1.3)e+04$ | $0.21 \pm 0.04$ |
| 6 | FCE7.1 | $9.0 \pm 0.6$ | 893 | $0.20 \pm 0.07$ | $196 \pm 31$ | $7.5 (\pm 2.0)e+04$ | $0.16 \pm 0.06$ |
| 7 | FCE11.2 | $6.1 \pm 0.1$ | 904 | $0.35 \pm 0.08$ | $153 \pm 31$ | $8.5 (\pm 2.1)e+04$ | $0.22 \pm 0.11$ |
| 8 | FCE11.3 | $7.2 \pm 0.1$ | 904 | $0.32 \pm 0.07$ | $172 \pm 14$ | $7.1 (\pm 2.9)e+04$ | $0.17 \pm 0.13$ |
| 9 | FCE13.3 | $9.4 \pm 0.1$ | 906 | $0.32 \pm 0.12$ | $230 \pm 38$ | $12.5 (\pm 4.6)e+04$ | $0.32 \pm 0.16$ |
| 10 | FCE22.0 | $-1.1 \pm 0.3$ | 899 | $0.26 \pm 0.05$ | -- | -- | -- |
| 11 | FCE22.1 | $1.3 \pm 0.1$ | 892 | $0.31 \pm 0.05$ | $238 \pm 21$ | $12.5 (\pm 1.7)e+04$ | $0.30 \pm 0.05$ |
| 12 | FCE24.0 | $-3.0 \pm 0.9$ | 907 | $0.14 \pm 0.05$ | -- | -- | |
| 13 | FCE26.1 | $2.6 \pm 0.4$ | 893 | $0.19 \pm 0.05$ | $213 \pm 53$ | $5.6 (\pm 0.8)e+04$ | $0.10 \pm 0.02$ |
| 14 | FCE26.2 | $1.4 \pm 0.4$ | 895 | $0.15 \pm 0.07$ | $152 \pm 25$ | $4.3 (\pm 1.3)e+04$ | $0.08 \pm 0.03$ |





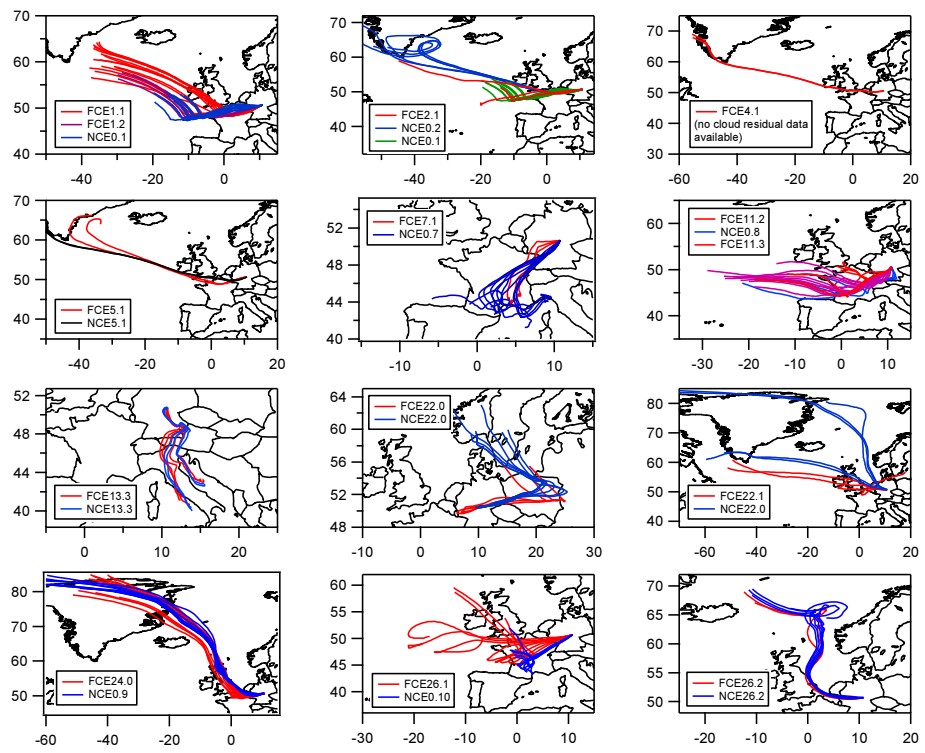

5    **Figure 1: Back trajectories calculated using HYSPLIT (Stein et al., 2015; Rolph, 2016) for all full cloud events (FCE) and the according cloud free periods. Details of the trajectory calculations are described in the supplement to Tilgner et al. (2014).**





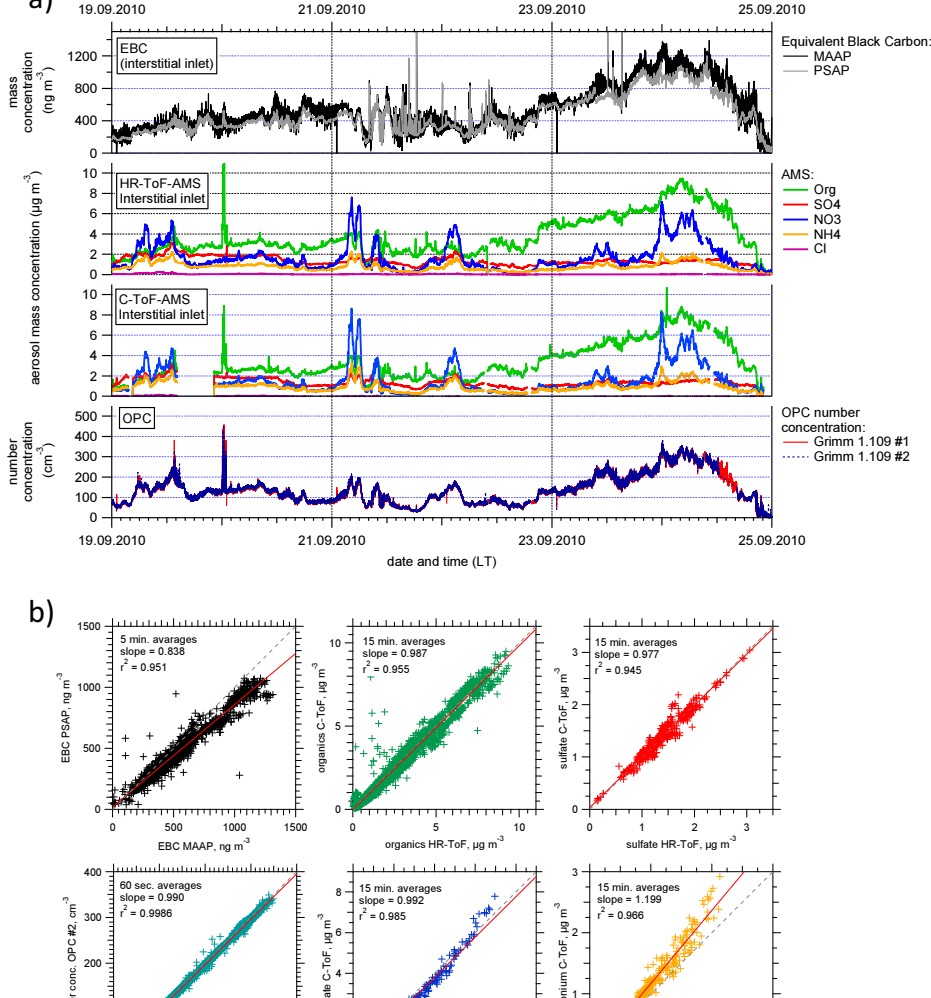

**Figure 2: a)** Time series of aerosol mass concentrations (EBC, organics, sulfate, nitrate, ammonium, and chloride) and number concentration ($d_p > 250$ nm) measured during 6 cloud-free days when all instruments sampled through the interstitial inlet. **b)** correlation plots of the compared data sets along with slopes and regression coefficients. The respective averaging times are indicated.





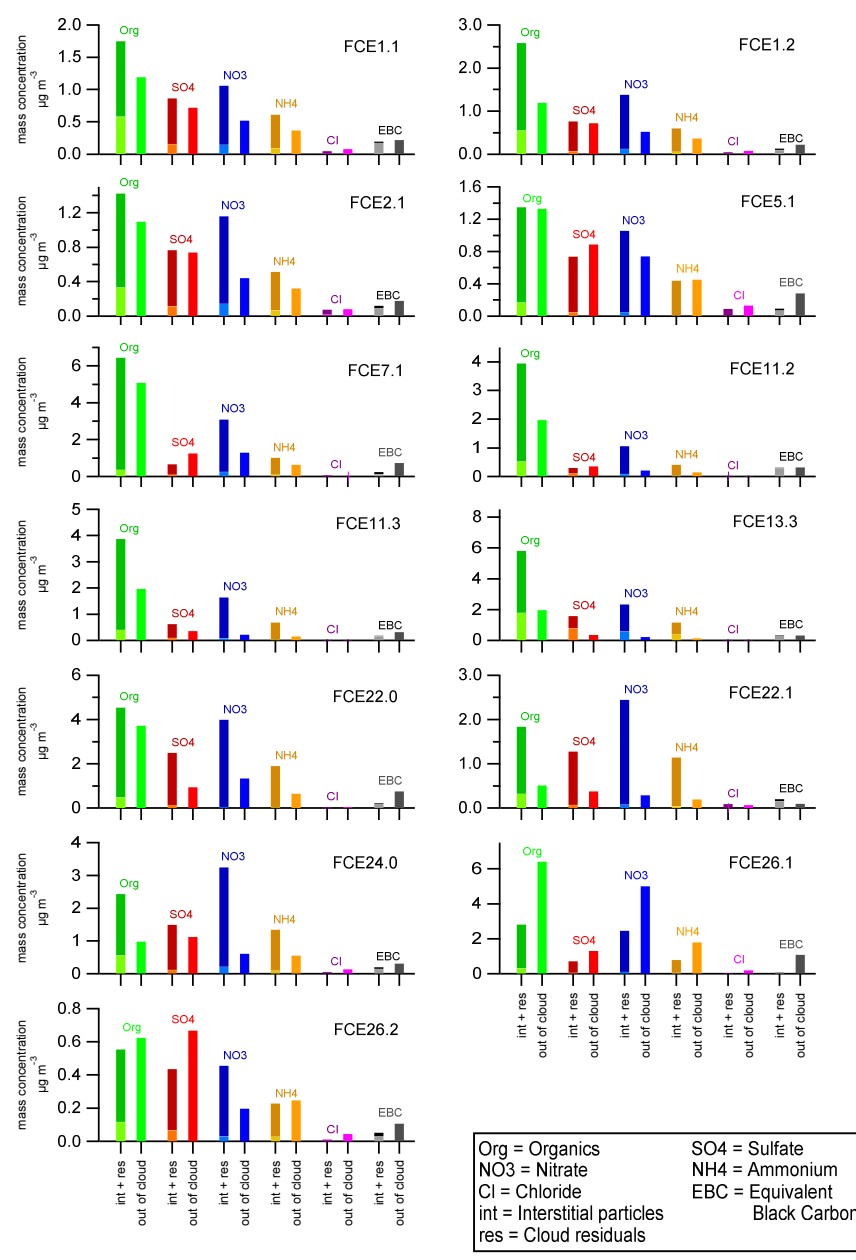

**Figure 3: Composition of cloud residual particles and interstitial aerosol during the full cloud events (FCE) and aerosol composition during corresponding non-cloud times. Interstitial and out-of-cloud aerosol was measured using the HR-ToF-AMS, cloud residuals were analyzed using the C-ToF-AMS.**



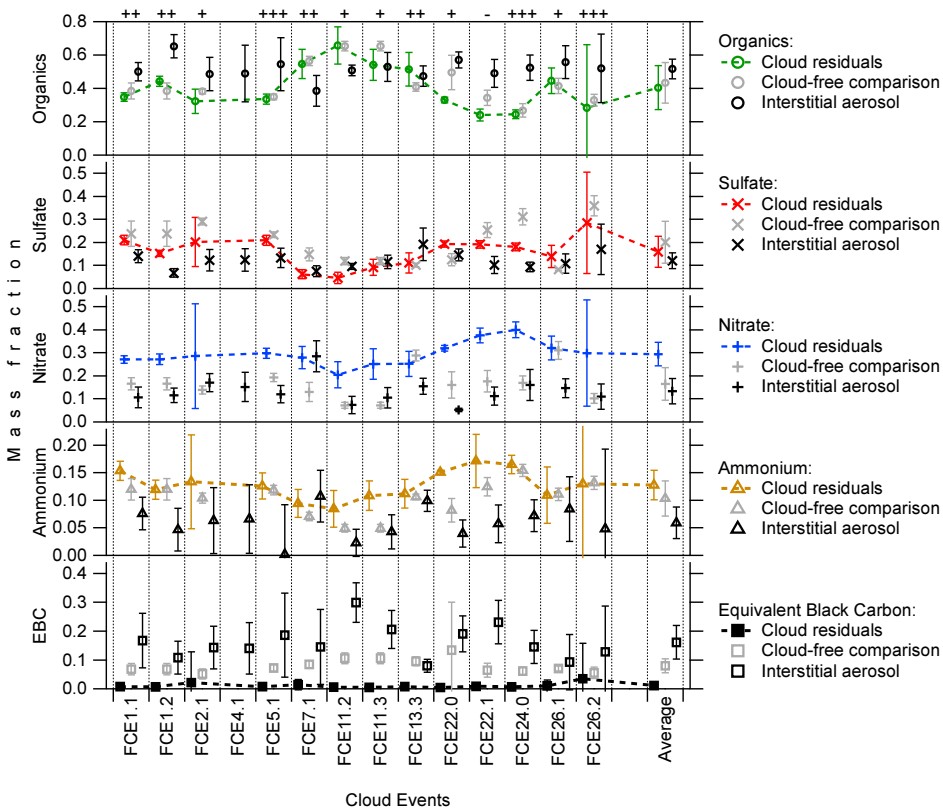

**Figure 4: Mass fractions of organics, sulfate, nitrate, ammonium, and EBC in the submicron aerosol, for cloud residuals, interstitial aerosol, and the cloud-free comparison periods. Error bars indicate standard deviations during the averaging period, thereby illustrating the variability during the cloud events. The similarity between the trajectories for cloud events and cloud-free periods from Figure 1 and Table 1 is indicated above the top graph.**





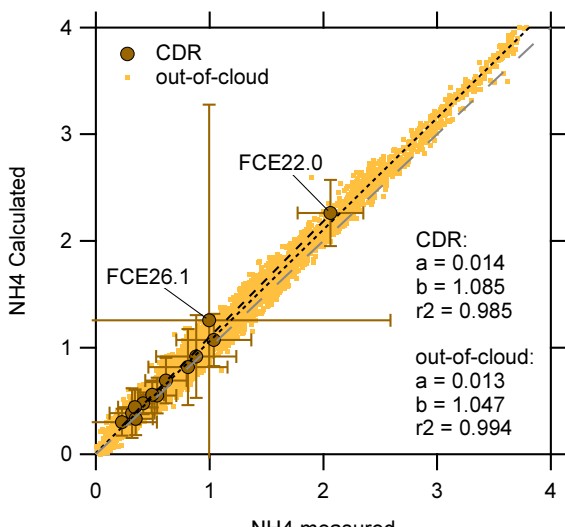

**Figure 5: Predicted ammonium (assuming full neutralization of nitrate and sulfate) versus measured ammonium, for cloud residuals (CDR) and out-of-cloud aerosol.**





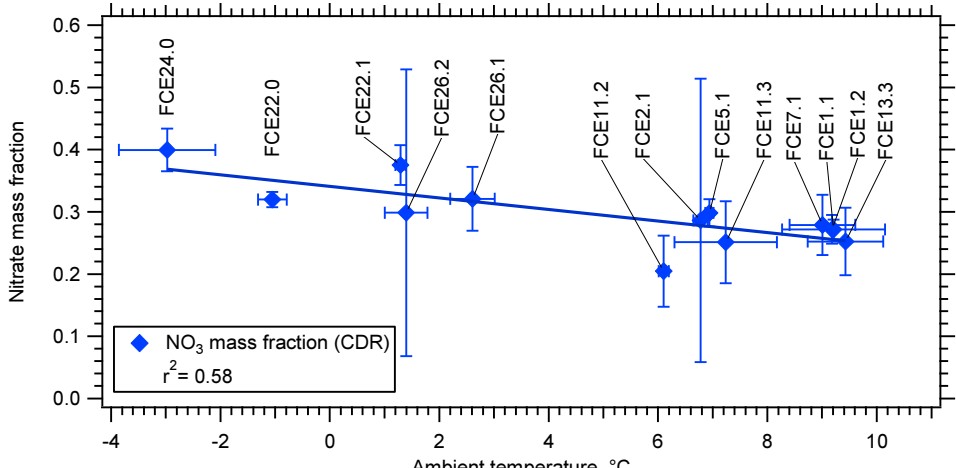

5  **Figure 6: Mass fraction of nitrate (mean value for each FCE) in cloud droplet residues (CDR) as function of temperature. Error bars represent the standard deviation of concentration and temperature during the cloud events. The line represents a linear fit to the data, the correlation coefficient is given in the legend.**



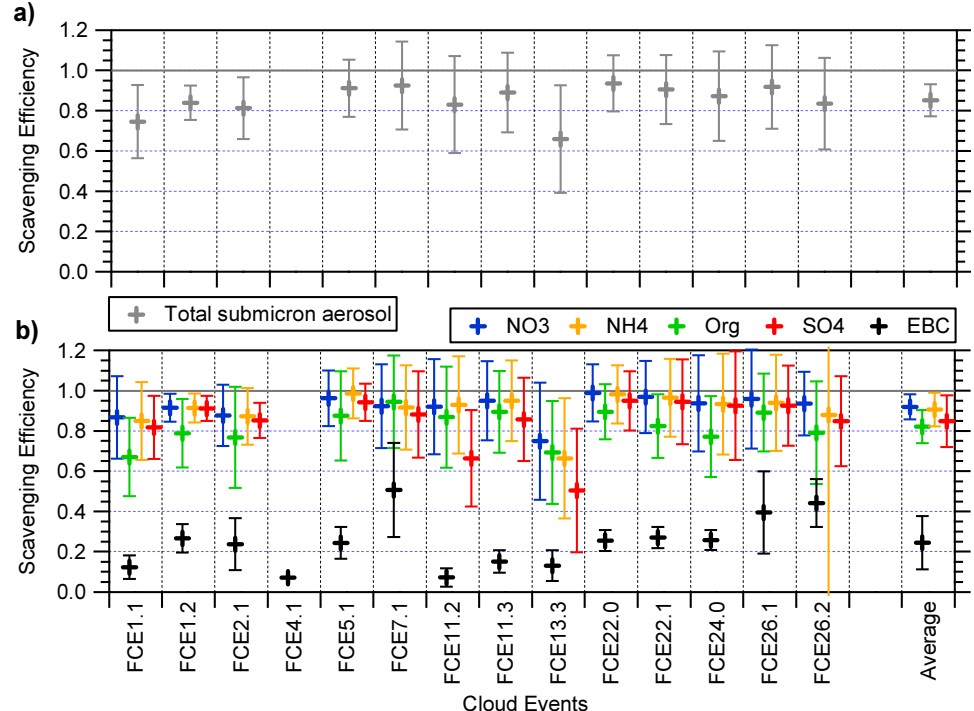

**Figure 7: Partitioning between cloud phase and interstitial phase: Scavenging efficiency** $SE = M_{cloud\ residuals}/(M_{cloud\ residuals} + M_{intersitital})$ **for the total aerosol mass (a) and the individual compounds (b). During FCE4.1, cloud residual data were not measured with the AMS. Error bars represent standard deviations of the time-resolved scavenging efficiencies during the individual FCE combined with the uncertainty of the CVI correction factors.**





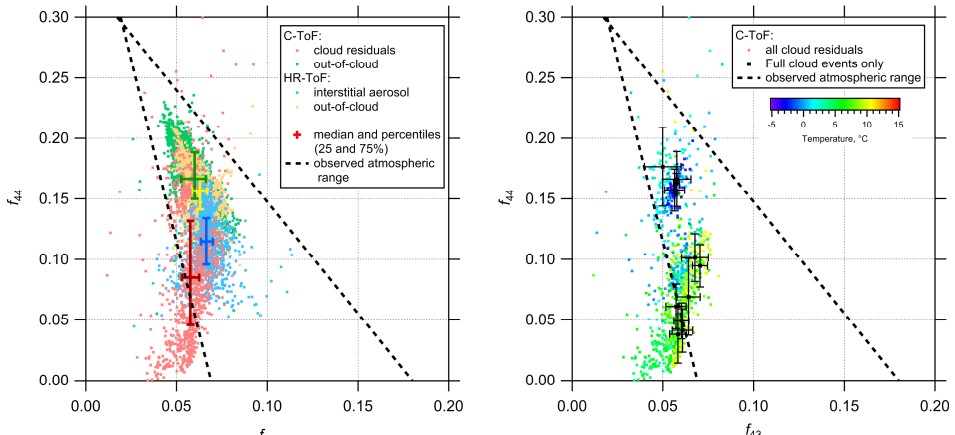

**Figure 8: Left:** $f_{44}$ **vs.** $f_{43}$ **for cloud residuals (red), interstitial aerosol (blue), and out-of-cloud aerosol (yellow and green) for the whole HCCT2010 data set. Each data point represents a 15-min average. Median and quartiles for all four data sets are given. The dotted lines denote the range of atmospheric values as reported by Ng et al. (2010). Right: Same plot, but only for cloud residues, colour coded by ambient temperature. The averaged values for the full cloud events are given by the black markers. Error bars represent standard deviation.**





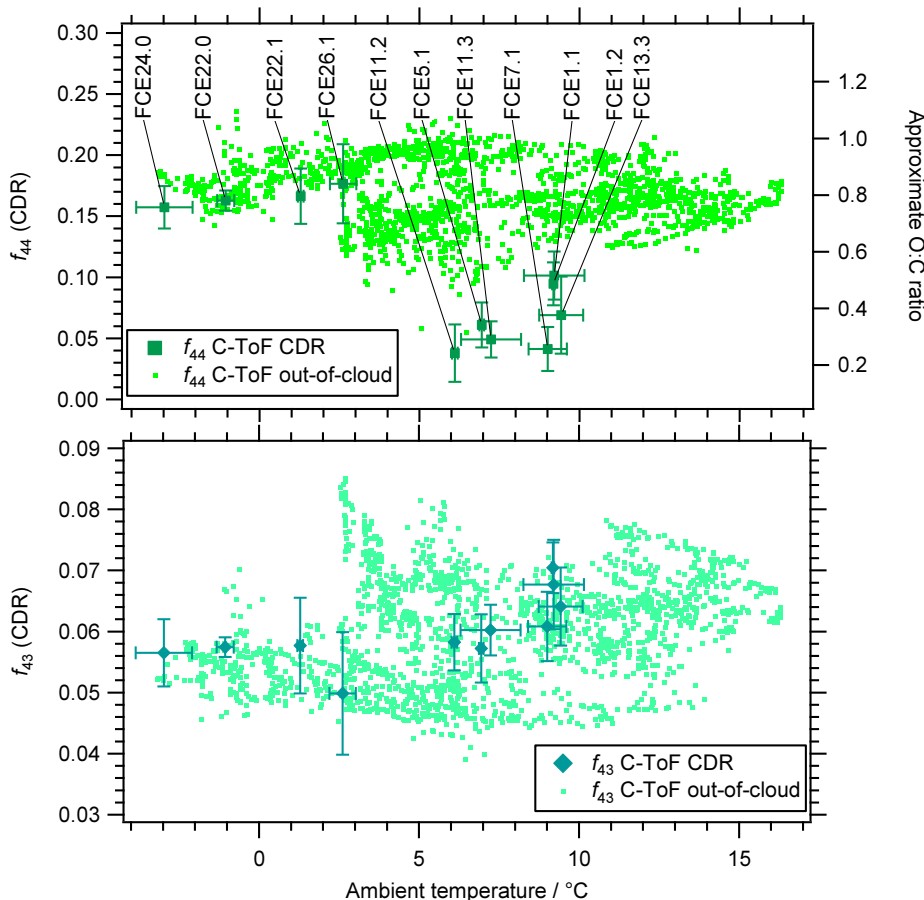

**Figure 9: Upper panel: Ratio m/z44 to total organics ($f_{44}$) of cloud droplet residues (CDR) and out-of-cloud aerosol (both measured with the C-ToF-AMS) as function of temperature. The approximate O:C ratio inferred from $f_{44}$ is given on the right axis. Lower panel: same graph but for $f_{43}$.**





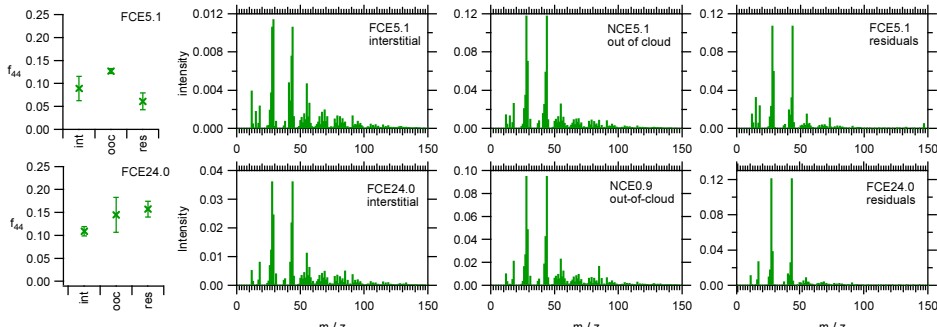

**Figure 10:** $f_{44}$ **values (with standard deviations) and organic mass spectra for interstitial aerosol, out-of-cloud aerosol, and cloud residuals, for two selected cloud events with low and high temperature (upper row: FCE5.1, 6.9°C; lower row: FCE24.0, -3.0°C) and best match between cloud and out-of-cloud trajectories (Table 1). Note that the** $f_{44}$ **values measured with the C-ToF-AMS have been scaled to the HR-ToF-AMS as described in Section 2.4 and Fig. S3, but the mass spectra are plotted as measured.**




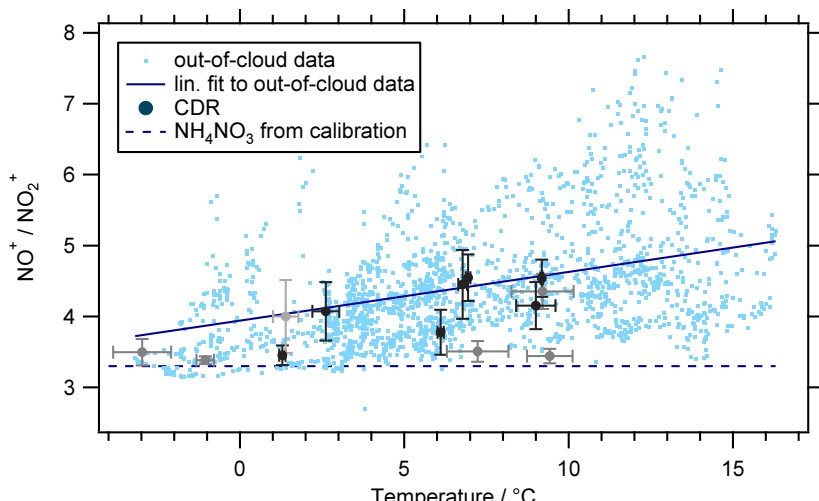

**Figure 11: Ratio of NO⁺ (m/z 30) to NO₂⁺ (m/z 46) measured using the C-ToF-AMS as a function of temperature. Out-of-cloud data represent 15min averages, CDR data are averaged over the full cloud events, error bars denote standard deviation. The solid line is a linear fit to the out-of-cloud data, the dashed line shows the ratio obtained during calibration with pure NH₄NO₃. The CDR data are color coded. Black: nighttime cloud events, dark grey: day and night, light grey: daytime event.**





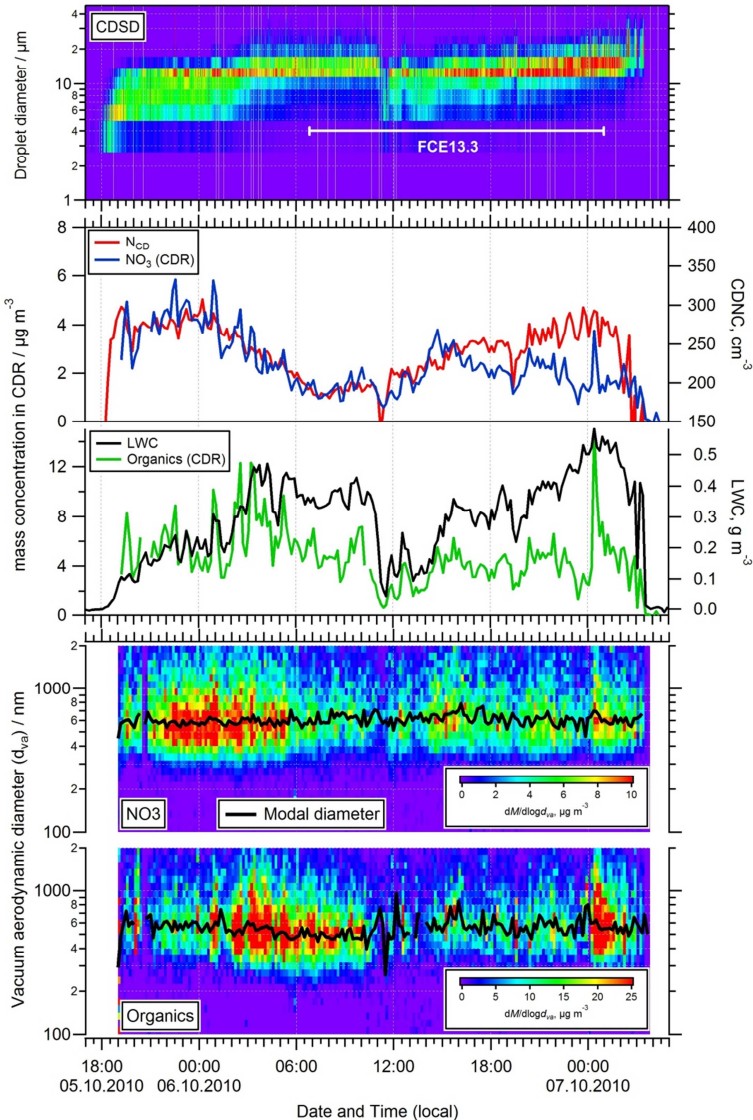

**Figure 12: Example of the temporal evolution of cloud and residue properties on 06 October 2010. Upper panel: Cloud droplet size distribution (CDSD) measured with the FSSP; Second panel: Cloud droplet number concentration (CDNC) measured with the FSSP along with nitrate mass concentration in cloud drop residues (CDR); Third panel: Liquid water content (LWC) measured by the PVM along with organic mass concentration in CDR; Forth and fifth panel: mass size distributions and modal diameter of nitrate and organics in CDR. All data represent 10min averages except for the CDSD which is given on a 5sec time base.**