# Peer review of "Uptake of nitric acid, ammonia, and organics in orographic clouds: Mass spectrometric analyses of droplet residual and interstitial aerosol particles"

_Atmospheric Chemistry and Physics, 2016_

## Referee Comment (RC1) · Anonymous Referee #1 · 28 Sep 2016

The study reports on a valuable dataset collected at the Schmucke mountain site in central Germany in September and October 2010. They collected aerosol composition data using a C-ToF-AMS downstream of a CVI inlet, which selectively samples droplets and rejects interstitial aerosol particles. The topic of the paper is of importance and of interest to readers of this journal. In general, measurements of this nature are difficult to make, especially at mountain sites, and thus the data are of importance to document in the literature. The analysis by the authors leads to a few interesting conclusions: (i) a significant fraction of submicron aerosol partitioned to the cloud liquid phase (85% on average); (ii) nitrate generally exhibited higher scavenging efficiencies

as compared to ammonium, sulfate, and organics (black carbon was the lowest); (iii) nitrate and ammonium mass fractions were enhanced in droplet residual particles, with a speculation made about temperature playing a role in this finding; (iv) the oxidation state of organic matter in droplet residuals was also shown to have a potential temperature dependence. I found the analysis to be supportive of the conclusions reached. The figures can benefit in some parts from better aesthetic quality, including larger font size. Figure 3 in particular could use improvement. The title of the work is supportive of the contents of the paper. I recommend publication of this work after the authors address my suggested minor revisions below. Most of the specific comments relate to incorporating the work of others that may have gone overlooked but are highly relevant to the discussion topics of this paper.

Specific Comments: Page 2, Line 33-36: Other papers have also shown this that should be mentioned: Asa-Awuku et al. (2015). CCN properties of organic aerosol collected below and within marine stratocumulus clouds near Monterey California, Atmosphere, 6, 1590-1607, doi:10.3390/atmos6111590.

Page 4, Lines 5-19: How hot does the interior of the CVI inlet become? Provide temperature information for the heated counterflow stream. Also, what are the flow rates used for the various streams of the inlet?

Page 5, Lines 23-27: Doesn't the sampling efficiency depend on the droplet size distribution in ambient air? If the droplet distribution is not held fixed, it seems as though some sizes may have better or worse sampling efficiencies. Discussion about this issue is warranted here.

Table 1: clarify in caption what is meant by the various numbers of "+" and "-" in the last column.

Figure 1: clarify how far back the trajectories go in time, and what the final ending coordinates and altitude are for the trajectories.

Figure 3: It is unclear how to read the bars. Specifically, what are the two shadings indicative of on the bars labeled "int+res"? It is very difficult to analyze the results in this figure due to the inability to understand that important bar. Caption and figure should be improved.

Page 8, Line 6: change "that" to "than"

Section 3.2.1: The authors should also incorporate into the discussion the recent results of a paper focused on this very issue: Prabhakar et al. (2014). Sources of nitrate in stratocumulus cloud water: Airborne measurements during the 2011 E-PEACE and 2013 NiCE studies, Atmos. Environ., 97, 166-173, doi:10.1016/j.atmosenv.2014.08.019. General comment: was there any evidence of influence from biomass burning in this study?

Page 13: Line 24-26: The authors should update their references here because more studies than they have listed have examined cloud residues using an AMS, with results that could be relevant to interpretation of their own results. Below are a few examples that should be included:

Coggon et al. (2014). Observations of continental biogenic impacts on marine aerosol and clouds off the coast of California, J. Geophys. Res., 119, doi:10.1002/2013JD021228.

Sorooshian et al. (2013). Observations of sharp oxalate reductions in stratocumulus clouds at variable altitudes: organic acid and metal measurements during the 2011 E-PEACE campaign, Environ. Sci. Technol., 47, 7747–7756, doi:10.1021/es4012383.

Coggon et al. (2012). Ship impacts on the marine atmosphere: Insights into the contribution of shipping emissions to the properties of marine aerosol and clouds, Atmos. Chem. Phys., 12, 8439-8458.

Wonaschuetz et al. (2012). Aerosol and gas re-distribution by shallow cumulus clouds: an investigation using airborne measurements, J. Geophys. Res., 117, D17202,

doi:10.1029/2012JD018089.

Shingler et al. (2012). Characterisation and airborne deployment of a new counterflow virtual impactor inlet, Atmos. Meas. Tech., 5, 1259–1269.

Sorooshian et al. (2007). Particulate organic acids and overall water-soluble aerosol composition measurements from the 2006 Gulf of Mexico Atmospheric Composition and Climate Study (GoMACCS), J. Geophys. Res., 112, D13201, doi:10.1029/2007JD008537.

Section 3.2.4: it may be worth mentioning that in a recent study (Below), an organonitrate species was found only in cloud water as compared to CDR and out-of-cloud aerosol owing to the effect of heat kicking the species out of the aerosol phase. The results of this particular study are indeed interesting and warrant future investigation as to the effect of temperature on CDR composition.

Youn et al. (2015). Dimethylamine as a major alkyl amine species in particles and cloud water: observations in semi-arid and coastal regions, Atmos. Environ., 122, 250-258, doi:10.1016/j.atmosenv.2015.09.061.

General comment: What are the key sources of nitric acid, ammonia, and organics in the region? Also, what about organonitrates? Please add discussion about this.

---

## Referee Comment (RC2) · Anonymous Referee #2 · 26 Oct 2016

Schneider et al. report the results of a field study aimed at studying the uptake of trace gases into clouds during the HCCT-2010 campaign in Thuringia, Germany. For this study two co-located aerosol mass spectrometers (AMS) measured the composition of residues from evaporated cloud droplets (i.e., residual particles) and interstitial aerosols during full cloud and cloud-free events. Aerosol species that form the focus of this manuscript include nitrate, ammonium, black carbon, organic carbon, and various (physical) aerosol and meteorological properties. The data accumulated during this campaign is quite unique among cloud composition studies. Only a handful of AMS studies exist related to the composition of cloud residues. This study appears to be the

first to measure interstitial aerosol and cloud droplets simultaneously using the two co-located AMS and their respective aerodynamic cutoff diameter and counterflow virtual impactor inlets.

The authors have collected a wealth of very high quality data from ∼14 full cloud events, in addition to non-cloud events for comparison. An extensive analysis of the data has been carried out that provides important insights into trace-gas-cloud and aerosol-cloud interactions. Very strong evidence, backed up by theoretical considerations and back trajectories, is provided showing that clouds effectively scavenge gas phase nitric acid, ammonia, volatile organic compounds, and less so black carbon. The data was used to calculate scavenging efficiencies for these gases, which range from 0.6 to 1.0 for nitric acid, ammonia, and VOCs, and 0.1 to 0.5 for black carbon. The methods employed are state-of-the science. The only data missing (due to lack of instrumentation at the campaign) were direct gas phase measurements of HNO3, H2SO4, NH3 and VOCs. However, what is missing in gas phase measurements is made up for by the thoroughly and cautiously interpreted AMS data, which leads to high confidence in the conclusions. I support publication in ACP after the few minor comments listed below are addressed.

Minor Comments:

The title includes the label of "orographic" clouds, but orographic does not appear again in the abstract or main text of the article. I recommend including in the text a description of the cloud type and what measured parameters defined the clouds as orographic during the campaign. Perhaps a discussion related to this could be added to section 3.1?

Molecular formulae should include subscripts on lines 19-21 on p. 2.

p. 4: Consider spelling out the abbreviations C-ToF-AMS and HR-ToF-AMS where they are first used.

[Figure]

p. 8, line17: Should read: "…relative to the sum of all species detected by the AMS…"

p. 8, line 23: replace "forth" with "fourth"

p. 9, line 25: replace "+" with "−" and visa versa for the charge on nitrate and ammonium, respectively.

p. 9, lines 27-31: The sentence beginning and ending in "If the air…occurs in the cloud phase," is a run-on sentence and is unclear. Please simplify or rephrase to improve clarity.

p. 14, line 23: Insert "show." Should read: "…our CDR data do not show such a clear trend…"

p. 16, line 32: Insert "with." Should read: "…was better correlated with CNDC than…"

p. 17, lines 21-23: The sentence beginning and ending with, "At the same temperature…the cloud is established," is unclear. Please rephrase.

Figure 9: I found the blue-green diamond symbols difficult to see over the background of green squares. I suggest changing the C-ToF CDR symbols to different colors so they stand out.

Figure 11: I find the different shades of grey difficult to tell apart in some print outs. Consider chaning colors of the CDR data
* * *

---

## Referee Comment (RC3) · Anonymous Referee #3 · 8 Nov 2016

This study describes aerosol chemical properties measured from both droplet residues and interstitial aerosol particles. The principal aims of this work are to study the cloud processing, including the enrichment of aerosol particles within clouds from the uptake of different gas-phase species. This paper also addresses the role of different chemical species in the activation of cloud droplets. The study is very thorough with an impressive instrumental setup and a large number of statistically relevant cloud events. The paper, figures, and text are well prepared. However, I have some major concerns regarding the experimental approaches used to derive the conclusions made in this manuscript.

[Figure]

General comments:

This experiment, the HCCT was intended to understand how aerosols are activated into clouds as well as the impact of different cloud processing on cloud properties. As outlined in Tilgner et al., (2014), this study was designed in such a way as to have three well equipped stations before, in, and after the formation of an orographic cloud. I would assume that combining measurements from these three stations would have made this study much more robust, rather than only comparing in and out of cloud residues on the cloud top.

1) One of my major critics is the lack of a clear discussion on the aerosol (CDR and interstitial) physical properties (size and number concentration). These factors play an essential role in the activation of cloud droplets and should not be separated from aerosol chemical properties. Aerosol size distributions should have been taken into account to provide a measure of aerosol activation diameter. It would have been interesting to investigate how this parameter (aerosol activation diameter) varied as a function of chemical composition. It would have been equally interesting to study the sampling efficiency of the CVI inlet through comparison of the total number of CDR particles (CPC/SMPS) with the total number of cloud droplets measured with the FSSP.

a. In section 2.3, the authors state that there are SMPS measurements available, however unless I am mistaken I do not find any other reference to these measurements, either in the manuscript or in the supplementary material.

b. Figure S6 shows OPC size distributions measured behind the CVI and Interstital inlet. The GRIMM instrument normally provides particle size distribution measurements from 300 nm up to > 10 microns. At 300 nm, all particles are expected to act as CCN. Therefore, these measurements are not useful to observe activation parameters of aerosol particles. I did not find any reference to this figure in the main manuscript.

2) The papers main results are based on the comparison of interstitial aerosol particles and cloud droplet residues. These two "types" of aerosol particles are found in largely

different size categories, with interstitial aerosol particles generally having diameters < 90 nm and CDR particles having diameters > 90 nm. It has been reported in a large number of studies that the contribution of organic aerosol particles increases as particles size decreases. Equally inorganic nitrates are often measured in larger particle diameters. Can the authors show that the increased organic compounds measured in the interstitial aerosol during cloud events are significant to the cloud event itself and that the concentrations (in the same size class) are different in the NCE?

a. Page 7, Line 35: The authors state that the mass concentration of the interstitial aerosol is lower than that of the CDR. This would be expected since CDR particles are larger in diameter (hence more mass) than the interstitial aerosol.

b. Figure 4. It would be useful to see the significance of the difference between the interstitial and CDR composition, through comparison of the similar size fractions (< 90 nm (INT) and > 90 nm (CDR)) during NCE.

3) Another concern is that the transmission efficiency of the aerodynamic lens used for Aerodyne products sample aerosol particles with "good" efficiency between ~90 nm and ~300 nm (Liu et al., 2007), however below (and above) these limits the transmission efficiency of the instrument decreases rapidly. One needs to take this transmission efficiency into account and also the implications that this may have on the quality of the AMS data at these lower diameters. Baseline errors will likely have an impact at these diameters so it is necessary to take precautions to ensure that measured aerosol compositions and concentrations at these small diameters are real and not just arbitrary noise.

Peter S. K. Liu, Rensheng Deng, Kenneth A. Smith, Leah R. Williams, John T. Jayne, Manjula R. Canagaratna, Kori Moore, Timothy B. Onasch, Douglas R. Worsnop, and Terry Deshler Transmission Efficiency of an Aerodynamic Focusing Lens System: Comparison of Model Calculations and Laboratory Measurements for the Aerodyne Aerosol Mass Spectrometer Aerosol Science And Technology Vol. 41 , Iss. 8,2007

4) A large part of this manuscript is focused on the enrichment of nitrate in aerosol particles after cloud processing. However, these conclusions were made through comparing CDR with NCE before, and in some cases, after the cloud event. If cloud processing was indeed used to result in the enrichment of nitrate in particles, would one not expect to observe higher nitrate in aerosol particles once the cloud event has passed?

5) A constant correction efficiency (CE) of 0.5 was applied to all data. However, there are several periods (shown in Figure 2 a)) where the contribution of nitrate aerosol particles was greater than 25% to the total aerosol mass. In general, within the aerosol mass spectrometry community, it is recommended to apply a composition dependent CE as outlined in the manuscript Middlebrook et al., (2012). Middlebrook,A.N R. Bahreini, J. L. Jimenez, and M. R. Canagaratna (2012) Aerosol Sci. Tech, 46:258–271.

6) Cloud events listed in table 1 varied from 3 hrs up to 12 hrs. Air mass trajectories were used to verify that there was no change in air mass properties, however could there be more robust criteria used to classify these cloud events. Could the authors incorporate the FSSP cloud droplet distribution and LWC measurements to evaluate whether the cloud properties changed outside a certain limit. For example Fig. 12 shows the cloud droplet diameters and concentrations changing during the cloud event, this was accompanied also by a change in the LWC. How can the authors ensure that these changes in cloud properties were not accompanied by slight air mass changes, or entrainment of new aerosol types. This might influence the comparison with NCE.

a. Likewise, how long a time period should be compared from the NCE data? It might not be judicious to include data from 24 hours prior to the measurements.

Minor comments: The only mention of orographic clouds is in the title of this manuscript. Although, full details of the experimental design is included in Tilgner et al., some discussion of the importance and how these cloud events were verified as orographic should be included.

Page 4, section "Analysis instruments" How was the aerosol dried prior to sampling in the interstitial aerosol?

Page 13, Line 33: The authors mention that biogenic emissions could be a source of the higher OA measured at higher temperatures. Is their any evidence of biogenic emissions during these periods? Gas-phase measurements, lower than average BC concentrations, etc.

Page 16, Line 29: What is the significance of these correlations? 99%, 95% and how is the 'significance' determined?
* * *

---

## Referee Comment (RC4) · Anonymous Referee #4 · 20 Nov 2016

This work provided comprehensive measurements during HCCT-2010 campaign at the Schmucke mountain site towards to understand the cloud properties. The simultaneous measurement of cloud droplets residues and interstitial aerosols with two AMS highlight the role of cloud processing in enriching aerosol particles by uptake of reactive gas species. This kind of dataset with high quality is rarely obtained and of great value. The analysis of temperature dependence of mass fraction of nitrate and distribution of f44 and f43 shone a light on the uptake/scavenging process of the chemicals, indicating the adverse effect of temperature on the uptake of nitric acid gases and more oxidized organics onto cloud droplets. Moreover, the organic nitrate concentration in

cloud droplet residual (CDR) is discussed as well. A positive correlation between organic nitrate in CDR and temperature was suggested. In addition, the authors have added a case study to further compare the effect of activation of nitrate containing particles and uptake of nitric acid gas onto cloud droplets and therefore making the whole article a thorough and logical analysis of the chemical composition evolution between CCN, precursor gases, interstitial aerosol and cloud droplet residual. In general, the conclusion was reasonably supported by the data and analysis. The manuscript was overall well written. I recommend this work can be published after some miner revision.

P.2 line 15-25 The formula of molecule shall be rewritten with subscripts.

P.3 line 30 Please clarify the "similarity" of trajectories. Is it subjective or did you use any objective method?

P.5 line 4 Please unify the expression of temperature unit ('-' or 'minus' ) according to the ACP writing instructions.

P.8 line 1-14 The map of trajectories in figure 1 are somehow changed, making it difficult to understand the exact properties of different trajectories like length of path or the location of polluted regions. Please offer some vertical description like the height of airmass's center. It may help to understand the uptake of gases if there were no precursor measurements.

P.8 line35 The conclusion drawn here shall be more careful, since if the fraction of nitrate in CDR was elevated, the out-of-cloud aerosol could have an elevated fraction of nitrate as well due to the evaporation or re-partitioning.

P.10 line 18, Considering that the uptake of nitric acid gas, ammonia and organics caused elevation of corresponding CDR compositions, the authors should be more careful to use the term of "xx% of total submicron aerosol mass partitioned in to cloud phase", which is very likely to lead to a misunderstanding that the CDR composition came all from aerosols.

P.14 line 15-20 The abundance of a certain chemical composition is not always equal to formation. Only by taking the ratio and the absolute concentration into consideration could tell one if there is formation of the specie. Therefore, the difference here might indicate different mechanisms or different form of organic nitrates.

P.14 line 24, Rephrase "our CDR data do not such a clear trend:".

P.16 line 30, A r2=0.32 could not be described as "significant linear correlation".

P.35 Figure 9, There is approximate of O:C on the right axis in the upper panel while it's absent in the lower panel.
* * *

---

## Author Comment (AC1) · 3 Jan 2017

**acp-2016-835**

**Reply to Reviewer #1**

This reply is formatted as follows:

**Reviewer's comments**

Authors' reply

**Changes to Text**

The study reports on a valuable dataset collected at the Schmucke mountain site in central Germany in September and October 2010. They collected aerosol composition data using a C-ToF-AMS downstream of a CVI inlet, which selectively samples droplets and rejects interstitial aerosol particles. The topic of the paper is of importance and of interest to readers of this journal. In general, measurements of this nature are difficult to make, especially at mountain sites, and thus the data are of importance to document in the literature. The analysis by the authors leads to a few interesting conclusions:

(i) a significant fraction of submicron aerosol partitioned to the cloud liquid phase (85% on average); (ii) nitrate generally exhibited higher scavenging efficiencies as compared to ammonium, sulfate, and organics (black carbon was the lowest); (iii) nitrate and ammonium mass fractions were enhanced in droplet residual particles, with a speculation made about temperature playing a role in this finding; (iv) the oxidation state of organic matter in droplet residuals was also shown to have a potential temperature dependence. I found the analysis to be supportive of the conclusions reached. The figures can benefit in some parts from better aesthetic quality, including larger font size. Figure 3 in particular could use improvement. The title of the work is supportive of the contents of the paper. I recommend publication of this work after the authors address my suggested minor revisions below. Most of the specific comments relate to incorporating the work of others that may have gone overlooked but are highly relevant to the discussion topics of this paper.

We thank the reviewer for this positive rating of our manuscript

Specific Comments: Page 2, Line 33-36: Other papers have also shown this that should be mentioned: Asa-Awuku et al. (2015). CCN properties of organic aerosol collected below and within marine stratocumulus clouds near Monterey California, Atmosphere, 6, 1590-1607, doi:10.3390/atmos6111590.

We were not aware of this publication and we included a reference to it.

Page 4, Lines 5-19: How hot does the interior of the CVI inlet become? Provide temperature information for the heated counterflow stream. Also, what are the flow rates used for the various streams of the inlet?

The counter flow is not actively heated. The counter flow is made up by the so-called supply flow. The supply flow is guided to the CVI inlet tip, where it is divided to the sample flow (sucked back to the instruments) and the counter flow (going out of the inlet). The supply flow is prepared from compressed air inside the lab, i.e. the counter flow as well as the CVI interior is at room temperature and therefore markedly warmer than the probed cloudy air. The supply flow and thus the sample flow is filtered and dried to a dew point below -40°C, which is the main reason for the evaporation of the droplets as soon as they have passed the counter flow. Typical flow rates have been 12 L/min for the supply flow, 10 L/min for the sample flow and 2 L/min for the counter flow.

Page 5, Lines 23-27: Doesn't the sampling efficiency depend on the droplet size distribution in ambient air? If the droplet distribution is not held fixed, it seems as though some sizes may have better or worse sampling efficiencies. Discussion about this issue is warranted here.

Concerning the relevant experimental sampling efficiency, two different issues have to be considered. The sampling efficiency of the used CVI inlet itself was determined in the lab and is described in Schwarzenboeck et al. (2000). The cut-off curve is rather sharp with a droplet transmission from 0 to 100% within 2  $\mu$ m. Thus the shape of the droplet size distribution has only a very small influence on the sampling efficiency. In these lab determinations of the cutoff curve all droplet trajectories are aligned to the CVI inlet orientation. This is the main reason why an aircraft-based CVI system typically has a 100% sampling efficiency above its lower cut-off diameter. In ground-based applications this is different. First, the CVI inlet needs to be installed inside a wind tunnel to achieve the required large wind velocities at the inlet tip. Second, a ground-based CVI is directed into the main prevailing wind direction. During HCCT2010, the CVI was centered in direction of the preferred wind sector (232°) for connected flow conditions (Tilgner et al., 2014). Thus, droplets with trajectories non-aligned to the wind tunnel inlet, due to horizontal and vertical wind fluctuations, will be lost. Moreover, there are droplets that make it into the wind tunnel but with still non-perfect aligned trajectories with respect to the CVI inlet. As a consequence these droplets are sampled but have contact with the inner surface of the CVI, so that the residual particle is lost. In order to account for these loss processes, the overall sampling efficiency of the CVI system is derived as explained in the text.

Table 1: clarify in caption what is meant by the various numbers of "+" and "-" in the last column.

We added an explanation to the table caption as follows:

The similarity between the trajectories for cloud events and non-cloud comparison times is given in the last column (+++: same air mass trajectories, ++: small deviations, +: large deviations, -: different air mass origin).

Figure 1: clarify how far back the trajectories go in time, and what the final ending coordinates and altitude are for the trajectories.

The trajectories go back 96 h in time, the final ending coordinates were 50.65N, 10.77E, at 500 m above model ground level. We added this information to the caption of Figure 1:

Figure 1: Back trajectories calculated using HYSPLIT (Stein et al., 2015; Rolph, 2016) for all full cloud events (FCE) and the according cloud free periods. The trajectories go back 96 h in time, the end point is 50.65N, 10.77E, at 500 m above model ground level. Details of the trajectory calculations are described in the supplement to Tilgner et al. (2014).

Figure 3: It is unclear how to read the bars. Specifically, what are the two shadings indicative of on the bars labeled "int+res"? It is very difficult to analyze the results in this figure due to the inability to understand that important bar. Caption and figure should be improved.

We improved the explanation given in the figure caption. The shadings refer to the interstitial and residual mass concentration which are stacked in order to compare the sum of interstitial mass concentration and residual mass concentration ("int" + "res") to the out-of-cloud aerosol. This is now explained in the caption as follows:

Figure 03: Composition of cloud residual and interstitial particle mass concentration during the full cloud events (FCE) and particle mass concentration during corresponding non-cloud

times. Interstitial and out-of-cloud aerosol particles were measured using the HR-ToF-AMS, cloud residuals were analyzed using the C-ToF-AMS. The first bar shows the sum of the residual concentration ("res") and the interstitial concentration ("int"); residual concentration (darker colour) is stacked on top of the interstitial concentration (lighter colour).

Page 8, Line 6: change "that" to "than"

changed

Section 3.2.1: The authors should also incorporate into the discussion the recent results of a paper focused on this very issue: Prabhakar et al. (2014). Sources of nitrate in stratocumulus cloud water: Airborne measurements during the 2011 E-PEACE and 2013 NiCE studies, Atmos. Environ., 97, 166-173, doi:10.1016/j.atmosenv.2014.08.019.

We added the following to section 3.2.1: In a more recent study, Prabhakar et al. (2014) concluded from aircraft-based measurements in clouds that dissolution of HNO3 in cloud drops and nucleation scavenging of NO3-containing particles both contributed to enhanced nitrate concentration measured in cloud residuals.

General comment: was there any evidence of influence from biomass burning in this study?

There was influence of biomass burning, this has been reported in Roth et al. (2016). AMS Data (Fig. 8 in Roth et al.) showed that at end of campaign up to 0.3  $\mu$ g m-3 were attributed to biomass burning organic aerosol (BBOA) (inferred from "poor man's PMF"), while total organics were about 5  $\mu$ g m-3 at that time. The single particle data presented in Roth et al. show that a large fraction of the particles (25-30%) both in the out-of-cloud aerosol and in the cloud residuals showed biomass burning signatures. However, the AMS data indicate that the mass concentration of BBOA was rather low: About 6% according to the "poor man's PMF" estimation in Roth et al.), and a very small contribution of f60 (marker for levoglucosan) in Figure 10 of our manuscript.

Page 13: Line 24-26: The authors should update their references here because more studies than they have listed have examined cloud residues using an AMS, with results that could be relevant to interpretation of their own results. Below are a few examples that should be included:

Coggon et al. (2014). Observations of continental biogenic impacts on marine aerosol and clouds off the coast of California, J. Geophys. Res., 119, doi:10.1002/2013JD021228.

Sorooshian et al. (2013). Observations of sharp oxalate reductions in stratocumulus clouds at variable altitudes: organic acid and metal measurements during the 2011 E-PEACE campaign, Environ. Sci. Technol., 47, 7747–7756, doi:10.1021/es4012383.

Coggon et al. (2012). Ship impacts on the marine atmosphere: Insights into the contribution of shipping emissions to the properties of marine aerosol and clouds, Atmos. Chem. Phys., 12, 8439-8458.

Wonaschuetz et al. (2012). Aerosol and gas re-distribution by shallow cumulus clouds: an investigation using airborne measurements, J. Geophys. Res., 117, D17202, doi:10.1029/2012JD018089.

Shingler et al. (2012). Characterisation and airborne deployment of a new counterflow virtual impactor inlet, Atmos. Meas. Tech., 5, 1259–1269.

Sorooshian et al. (2007). Particulate organic acids and overall water-soluble aerosol composition measurements from the 2006 Gulf of Mexico Atmospheric Composition and Climate Study (GoMACCS), J. Geophys. Res., 112, D13201, doi:10.1029/2007JD008537.

We included these references to the discussion in section 3.2.3

Section 3.2.4: it may be worth mentioning that in a recent study (Below), an organonitrate species was found only in cloud water as compared to CDR and out-of-cloud aerosol owing to the effect of heat kicking the species out of the aerosol phase. The results of this particular study are indeed interesting and warrant future investigation as to the effect of temperature on CDR composition.

Youn et al. (2015). Dimethylamine as a major alkyl amine species in particles and cloud water: observations in semi-arid and coastal regions, Atmos. Environ., 122, 250-258, doi:10.1016/j.atmosenv.2015.09.061.

This is an interesting aspect. In our single particle paper (Roth et al., 2016) from the HCCT project we reported that amine-containing particles were detected with enhanced abundance in cloud residues. However, DMA is not an organonitrate (but an alkyl amine), thus we think that this would be off-topic here.

General comment: What are the key sources of nitric acid, ammonia, and organics in the region? Also, what about organonitrates? Please add discussion about this.

There are no large cities in the region, especially in the upwind direction (see list of cities within a 50 km radius around the measurement site in the supplement to Roth et al. (2016)). Thus, we expect the sources of organics and ammonia to be mainly from natural, biogenic sources, whereas nitric acid is most likely from NOx emissions (traffic or other) distributed over longer distances.

**References**

- Prabhakar, G., Ervens, B., Wang, Z., Maudlin, L. C., Coggon, M. M., Jonsson, H. H., Seinfeld, J. H., and Sorooshian, A.: Sources of nitrate in stratocumulus cloud water: Airborne measurements during the 2011 E-PEACE and 2013 NiCE studies, Atmospheric Environment, 97, 166-173, doi: 10.1016/j.atmosenv.2014.08.019, 2014.
- Rolph, G. D.: Real-time Environmental Applications and Display sYstem (READY) Website (http://ready.arl.noaa.gov), NOAA Air Resources Laboratory, Silver Spring, MD (last access: June 13, 2016), 2016.
- Roth, A., Schneider, J., Klimach, T., Mertes, S., van Pinxteren, D., Herrmann, H., and Borrmann, S.: Aerosol properties, source identification, and cloud processing in orographic clouds measured by single particle mass spectrometry on a central European mountain site during HCCT-2010, Atmos. Chem. Phys., 16, 505-524, doi: 10.5194/acp-16-505-2016, 2016.
- Schwarzenboeck, A., Heintzenberg, J., and Mertes, S.: Incorporation of aerosol particles between 25 and 850 nm into cloud elements: measurements with a new complementary sampling system, Atmos. Res., 52, 241-260, 2000.
- Stein, A. F., Draxler, R. R., Rolph, G. D., Stunder, B. J. B., Cohen, M. D., and Ngan, F.: NOAA's HYSPLIT Atmospheric Transport and Dispersion Modeling System, Bulletin of the American Meteorological Society, 96, 2059-2077, doi: doi:10.1175/BAMS-D-14-00110.1, 2015.
- Tilgner, A., Schöne, L., Bräuer, P., van Pinxteren, D., Hoffmann, E., Spindler, G., Styler, S. A., Mertes, S., Birmili, W., Otto, R., Merkel, M., Weinhold, K., Wiedensohler, A., Deneke, H., Schrödner, R., Wolke, R., Schneider, J., Haunold, W., Engel, A., Wéber, A., and Herrmann, H.:

Comprehensive assessment of meteorological conditions and airflow connectivity during HCCT-2010, Atmos. Chem. Phys., 14, 9105-9128, doi: 10.5194/acp-14-9105-2014, 2014.

---

## Author Comment (AC2) · 3 Jan 2017

**acp-2016-835**

**Reply to Reviewer #2**

This reply is formatted as follows:

Reviewer's comments

>  Authors' reply

>  Changes to Text

Schneider et al. report the results of a field study aimed at studying the uptake of trace gases into clouds during the HCCT-2010 campaign in Thuringia, Germany. For this study two co-located aerosol mass spectrometers (AMS) measured the composition of residues from evaporated cloud droplets (i.e., residual particles) and interstitial aerosols during full cloud and cloud-free events. Aerosol species that form the focus of this manuscript include nitrate, ammonium, black carbon, organic carbon, and various (physical) aerosol and meteorological properties. The data accumulated during this campaign is quite unique among cloud composition studies. Only a handful of AMS studies exist related to the composition of cloud residues. This study appears to be the first to measure interstitial aerosol and cloud droplets simultaneously using the two collocated AMS and their respective aerodynamic cutoff diameter and counterflow virtual impactor inlets.

The authors have collected a wealth of very high quality data from ~14 full cloud events, in addition to non-cloud events for comparison. An extensive analysis of the data has been carried out that provides important insights into trace-gas-cloud and aerosol cloud interactions. Very strong evidence, backed up by theoretical considerations and back trajectories, is provided showing that clouds effectively scavenge gas phase nitric acid, ammonia, volatile organic compounds, and less so black carbon. The data was used to calculate scavenging efficiencies for these gases, which range from 0.6 to 1.0 for nitric acid, ammonia, and VOCs, and 0.1 to 0.5 for black carbon. The methods employed are state-of-the science. The only data missing (due to lack of instrumentation at the campaign) were direct gas phase measurements of HNO3, H2SO4, NH3 and VOCs. However, what is missing in gas phase measurements is made up for by the thoroughly and cautiously interpreted AMS data, which leads to high confidence in the conclusions. I support publication in ACP after the few minor comments listed below are addressed.

>  We thank the reviewer for this very positive rating of our manuscript

Minor Comments:

The title includes the label of "orographic" clouds, but orographic does not appear again in the abstract or main text of the article. I recommend including in the text a description of the cloud type and what measured parameters defined the clouds as orographic during the campaign. Perhaps a discussion related to this could be added to section 3.1?

>  A description of the cloud types is given in detail in other publications in the special issue on the HCCT campaign, e.g., Tilgner et al. (2014) or Roth et al. (2016). Therefore, we added only a brief description of the cloud to section 3.1.:

>  The whole time series of the cloud droplet number distribution measured by the FSSP is given in Figure S4. The 14 FCE are also indicated by the grey bars. These FCE were chosen based on certain criteria as detailed in Tilgner et al. (2014), focusing on connected flow conditions between the upwind, the summit and the downwind station. These conditions went along with stable south-west flow conditions, thus the clouds were mainly of orographic nature, however in certain cases the meteorological analysis revealed that the clouds were not purely

orographic (FCE1.1, FCE2.1, FCE26.1). For details see Table 5 in Tilgner et al. (2014) and Table 1 in Roth et al. (2016).

Molecular formulae should include subscripts on lines 19-21 on p. 2.

> This was corrected

p. 4: Consider spelling out the abbreviations C-ToF-AMS and HR-ToF-AMS where they are first used.

> Done.

p. 8, line17: Should read: "…relative to the sum of all species detected by the AMS…"

> Changed

p. 8, line 23: replace "forth" with "fourth"

> Changed

p. 9, line 25: replace "+" with "–" and visa versa for the charge on nitrate and ammonium, respectively.

> This was corrected

p. 9, lines 27-31: The sentence beginning and ending in "If the air…occurs in the cloud phase," is a run-on sentence and is unclear. Please simplify or rephrase to improve clarity.

> We changed the sentence to:
>
> If no chemical processing of nitrate and ammonium occurs in the cloud phase and the air returns to the same temperature and relative humidity conditions after the cloud as before the cloud, is it to expect that the overall equilibrium between particle phase $NH_4NO_3$ and gas phase $NH_3$ and $HNO_3$ after the cloud will be equal to that before the cloud.

p. 14, line 23: Insert "show." Should read: "…our CDR data do not show such a clear trend…"

> Corrected

p. 16, line 32: Insert "with." Should read: "…was better correlated with CNDC than…"

> Corrected

p. 17, lines 21-23: The sentence beginning and ending with, "At the same temperature…the cloud is established," is unclear. Please rephrase.

> We changed the sentence to:
>
> If the aerosol experiences the same temperature and relative humidity after the cloud passage as before the cloud passage, the same equilibrium between particle-phase ammonium nitrate and gas phase nitric acid and ammonia is established.

Figure 9: I found the blue-green diamond symbols difficult to see over the background of green squares. I suggest changing the C-ToF CDR symbols to different colors so they stand out.

We changed the CDR symbols in both panels to black.

Figure 11: I find the different shades of grey difficult to tell apart in some print outs. Consider chaning colors of the CDR data

We changed the colors of the CDR data to black, purple and red

References:

Roth, A., Schneider, J., Klimach, T., Mertes, S., van Pinxteren, D., Herrmann, H., and Borrmann, S.: Aerosol properties, source identification, and cloud processing in orographic clouds measured by single particle mass spectrometry on a central European mountain site during HCCT-2010, Atmos. Chem. Phys., 16, 505-524, doi: 10.5194/acp-16-505-2016, 2016.

Tilgner, A., Schöne, L., Bräuer, P., van Pinxteren, D., Hoffmann, E., Spindler, G., Styler, S. A., Mertes, S., Birmili, W., Otto, R., Merkel, M., Weinhold, K., Wiedensohler, A., Deneke, H., Schrödner, R., Wolke, R., Schneider, J., Haunold, W., Engel, A., Wéber, A., and Herrmann, H.: Comprehensive assessment of meteorological conditions and airflow connectivity during HCCT-2010, Atmos. Chem. Phys., 14, 9105-9128, doi: 10.5194/acp-14-9105-2014, 2014.

---

## Author Comment (AC3) · 3 Jan 2017

**acp-2016-835**

This reply is formatted as follows:

**Reply to Reviewer #4**

Reviewer's comments

      Authors' reply

      Changes to Text

This work provided comprehensive measurements during HCCT-2010 campaign at the Schmucke mountain site towards to understand the cloud properties. The simultaneous measurement of cloud droplets residues and interstitial aerosols with two AMS highlight the role of cloud processing in enriching aerosol particles by uptake of reactive gas species. This kind of dataset with high quality is rarely obtained and of great value. The analysis of temperature dependence of mass fraction of nitrate and distribution of f44 and f43 shone a light on the uptake/scavenging process of the chemicals, indicating the adverse effect of temperature on the uptake of nitric acid gases and more oxidized organics onto cloud droplets. Moreover, the organic nitrate concentration in cloud droplet residual (CDR) is discussed as well. A positive correlation between organic nitrate in CDR and temperature was suggested. In addition, the authors have added a case study to further compare the effect of activation of nitrate containing particles and uptake of nitric acid gas onto cloud droplets and therefore making the whole article a thorough and logical analysis of the chemical composition evolution between CCN, precursor gases, interstitial aerosol and cloud droplet residual. In general, the conclusion was reasonably supported by the data and analysis. The manuscript was overall well written. I recommend this work can be published after some miner revision.

      We thank the reviewer for this positive rating of our manuscript

P.2 line 15-25 The formula of molecule shall be rewritten with subscripts.

      Corrected.

P.3 line 30 Please clarify the "similarity" of trajectories. Is it subjective or did you use any objective method?

      The similarity was estimated by subjective inspection of the graphs shown in Figure 1

P.5 line 4 Please unify the expression of temperature unit ('-' or 'minus' ) according to the ACP writing instructions.

      Changed.

P.8 line 1-14 The map of trajectories in figure 1 are somehow changed, making it difficult to understand the exact properties of different trajectories like length of path or the location of polluted regions. Please offer some vertical description like the height of airmass's center. It may help to understand the uptake of gases if there were no precursor measurements.

      We added the pressure history of the air masses to Figure 1. We tried to unify the latitude and longitude scaling of the maps as good as possible, without losing information for those air masses that had traveled only shorter distances during the 96 hours.

P.8 line35 The conclusion drawn here shall be more careful, since if the fraction of nitrate in CDR was elevated, the out-of-cloud aerosol could have an elevated fraction of nitrate as well due to the evaporation or re-partitioning.

Our data show that in general the nitrate mass fraction is highest in the cloud residuals, higher than in the out-of-cloud-aerosol, and higher than in the interstitial aerosol. It is very likely that the out-of-cloud aerosol is influenced by previous cloud passages and may therefore contain more nitrate than in earlier times, but still our observations show that CDR have the highest nitrate fraction. A depletion of the nitrate fraction in the interstitial particles was not observed. Thus, to our opinion the only explanation for the elevated ammonium nitrate in the CDR remains uptake of gaseous HNO3 and NH3.

P.10 line 18, Considering that the uptake of nitric acid gas, ammonia and organics caused elevation of corresponding CDR compositions, the authors should be more careful to use the term of "xx% of total submicron aerosol mass partitioned in to cloud phase", which is very likely to lead to a misunderstanding that the CDR composition came all from aerosols.

This is a good point. We reformulated the statement to:

Figure 7 shows that on average 85 % of the total submicron aerosol mass is present in the cloud phase, with a maximum value of 94 % in FCE22.0 and minimum values of about 66 % in FCE13.3. This can be the result of three different processes: activation of the pre-existing aerosol particles acting as CCN, scavenging of interstitial, non-activated aerosol particles, and uptake of gas-phase species by the cloud droplets as it was discussed for nitrate and ammonium above.

P.14 line 15-20 The abundance of a certain chemical composition is not always equal to formation. Only by taking the ratio and the absolute concentration into consideration could tell one if there is formation of the specie. Therefore, the difference here might indicate different mechanisms or different form of organic nitrates.

We agree with that. Our observations suggest that the abundance of organic nitrate in the particle phase increases with temperature in our data set. We can't conclude from this observation to the formation mechanism. Therefore we reformulated the paragraph to:

This finding suggests that the abundance of organic nitrate in the particle phase increases with temperature. However, recent observations by Lee et al. (2014) showed that formation of organic nitrate is enhanced at lower temperatures. Since also the equilibrium between particle phase and gas phase should be shifted towards the gas phase at higher temperatures, the observed higher amount of organic nitrates in the aerosol phase at higher temperatures cannot be explained by the formation mechanism proposed by Lee et al. (2014).

P.14 line 24, Rephrase "our CDR data do not such a clear trend:".

Changed to: "our CDR data do not show such a clear trend"

P.16 line 30, A r2=0.32 could not be described as "significant linear correlation".

Significance was tested using both t-test and F-test (using Wavemetric's IGOR function "statslinearregression") with 95% confidence interval. The correlation of both, NO3 and NH4 versus temperature is significant (the regression coefficient is different from zero with 95% confidence).

P.35 Figure 9, There is approximate of O:C on the right axis in the upper panel while it's absent in the lower panel.

This is on purpose, because the O:C values were derived from the $f_{44}$ value (left scale in the upper panel) using the parameterization given in Aiken et al. (2008) and Canagaratna et al. (2015) (as explained in section 3.2.3), whereas this can not be done using the $f_{43}$ value (left scale in lower panel).

**References**

Aiken, A. C., Decarlo, P. F., Kroll, J. H., Worsnop, D. R., Huffman, J. A., Docherty, K. S., Ulbrich, I. M., Mohr, C., Kimmel, J. R., Sueper, D., Sun, Y., Zhang, Q., Trimborn, A., Northway, M., Ziemann, P. J., Canagaratna, M. R., Onasch, T. B., Alfarra, M. R., Prevot, A. S. H., Dommen, J., Duplissy, J., Metzger, A., Baltensperger, U., and Jimenez, J. L.: O/C and OM/OC ratios of primary, secondary, and ambient organic aerosols with high-resolution time-of-flight aerosol mass spectrometry, Environ. Sci. Technol., 42, 4478-4485, doi: 10.1021/es703009q, 2008.

Canagaratna, M. R., Jimenez, J. L., Kroll, J. H., Chen, Q., Kessler, S. H., Massoli, P., Hildebrandt Ruiz, L., Fortner, E., Williams, L. R., Wilson, K. R., Surratt, J. D., Donahue, N. M., Jayne, J. T., and Worsnop, D. R.: Elemental ratio measurements of organic compounds using aerosol mass spectrometry: characterization, improved calibration, and implications, Atmos. Chem. Phys., 15, 253-272, doi: 10.5194/acp-15-253-2015, 2015.

Lee, L., Wooldridge, P. J., Gilman, J. B., Warneke, C., de Gouw, J., and Cohen, R. C.: Low temperatures enhance organic nitrate formation: evidence from observations in the 2012 Uintah Basin Winter Ozone Study, Atmos. Chem. Phys., 14, 12441-12454, doi: 10.5194/acp-14-12441-2014, 2014.

---

## Author Comment (AC4) · 4 Jan 2017

**acp-2016-835**

**Reply to Reviewer #3**

This reply is formatted as follows:

Reviewer's comments

Authors' reply

Changes to Text

This study describes aerosol chemical properties measured from both droplet residues and interstitial aerosol particles. The principal aims of this work are to study the cloud processing, including the enrichment of aerosol particles within clouds from the uptake of different gas-phase species. This paper also addresses the role of different chemical species in the activation of cloud droplets. The study is very thorough with an impressive instrumental setup and a large number of statistically relevant cloud events. The paper, figures, and text are well prepared. However, I have some major concerns regarding the experimental approaches used to derive the conclusions made in this manuscript.

General comments:
This experiment, the HCCT was intended to understand how aerosols are activated into clouds as well as the impact of different cloud processing on cloud properties. As outlined in Tilgner et al., (2014), this study was designed in such a way as to have three well equipped stations before, in, and after the formation of an orographic cloud. I would assume that combining measurements from these three stations would have made this study much more robust, rather than only comparing in and out of cloud residues on the cloud top.

> It is correct that three field sites were operational during HCCT, one upwind, one downwind, and one summit site. In the ACP special issue ([http://www.atmos-chem-phys.net/special_issue287.html](http://www.atmos-chem-phys.net/special_issue287.html)) already 11 papers are published dealing with various aspects of the study from all three stations. A paper focusing on comparison between upwind and downwind site, especially with respect to aerosol composition, is currently under preparation. The purpose of the study presented here is to compare cloud residual composition with simultaneously measured interstitial particle composition, which can be done only at the summit site. Additionally, we chose to include out-of-cloud data measured at the same site, in order to minimize uncertainties arising from instrumental differences that we would have to face when comparing the data from the upwind and downwind station to the summit site data.

1) One of my major critics is the lack of a clear discussion on the aerosol (CDR and interstitial) physical properties (size and number concentration). These factors play an essential role in the activation of cloud droplets and should not be separated from aerosol chemical properties. Aerosol size distributions should have been taken into account to provide a measure of aerosol activation diameter. It would have been interesting to investigate how this parameter (aerosol activation diameter) varied as a function of chemical composition. It would have been equally interesting to study the sampling efficiency of the CVI inlet through comparison of the total number of CDR particles (CPC/SMPS) with the total number of cloud droplets measured with the FSSP.

> The focus of the present paper is not the activation of aerosol particle to cloud droplets. We focus on uptake of gaseous compounds (nitrate, ammonium and organics) by cloud droplets. We fully agree that activation diameter as a function of chemical composition is a very interesting topic and can be achieved by using the available SMPS data, but this is outside the scope of this paper. The size-resolved AMS data can not be used for such an analysis due to the low duty cycle of the size-resolved measurement mode of the AMS (see below). Sizesegregated cloud condensation nuclei (CCN) measurements from upwind and downwind station have been published by Henning et al. (2014), and particle hygroscopicity and CCN activity at the upwind site have been compared to their chemical composition by Wu et al. (2013).

The CVI sampling efficiency has been studied exactly as suggested by the reviewer, by comparing the number of residual particles counted behind the CVI (using a CPC) and the number of cloud droplets measured outside (using the FSSP).
This is already described in section 2.4 of our manuscript, but will add the FSSP and CPC for clarity:

> "The enrichment factor of the CVI is given by the ratio of the air flow in the CVI wind-tunnel to the sample flow inside the CVI inlet. Since both quantities are measured, the enrichment factor can be calculated. The sampling efficiency of the CVI is determined by comparing the number of residual particles counted by a CPC behind the CVI and the number of cloud droplets measured outside by the FSSP and by comparing the LWC measured in the CVI sampling line and the LWC measured outside. Both the enrichment factor and sampling efficiency were provided as a function of time and have been applied to the data presented here."

See also reply to reviewer #1 on the question to the sampling efficiency.

a. In section 2.3, the authors state that there are SMPS measurements available, however unless I am mistaken I do not find any other reference to these measurements, either in the manuscript or in the supplementary material.

> SMPS measurements have been conducted during HCCT by TROPOS, but these data are not subject of the present manuscript. We removed the SMPS from that sentence in section 2.3:
>
> Particle size distributions of the interstitial aerosol particles and the cloud residuals were measured using optical particle counters (OPC, model 1.109 and 1.108, Grimm Aerosol Technik, Germany).

b. Figure S6 shows OPC size distributions measured behind the CVI and Interstital inlet. The GRIMM instrument normally provides particle size distribution measurements from 300 nm up to > 10 microns. At 300 nm, all particles are expected to act as CCN. Therefore, these measurements are not useful to observe activation parameters of aerosol particles. I did not find any reference to this figure in the main manuscript.

> We added the OPC size distribution for completeness to the supplement. We agree that the OPC data can't be used to observe activation parameters, but, as said before, this is not the topic of the present manuscript.

2) The papers main results are based on the comparison of interstitial aerosol particles and cloud droplet residues. These two "types" of aerosol particles are found in largely different size categories, with interstitial aerosol particles generally having diameters < 90 nm and CDR particles having diameters > 90 nm. It has been reported in a large number of studies that the contribution of organic aerosol particles increases as particles size decreases. Equally inorganic nitrates are often measured in larger particle diameters. Can the authors show that the increased organic compounds measured in the interstitial aerosol during cloud events are significant to the cloud event itself and that the concentrations (in the same size class) are different in the NCE?

> The finding that organics are large in the interstitial is likely just due to the effect that the reviewer describes: Organics have larger mass fraction in smaller particles which are less effectively activated as CCN. It has been reported that "Size matters more than chemistry for

cloud-nucleating ability of aerosol particles" (Dusek et al., 2006). However, the lower hygroscopicity of organic compounds may play an additional role and lead to less activation of particles in the size range around and slightly above 90 nm when they are have a higher organic content. But generally one would not expect that the interstitial aerosol composition is much different from the composition of the out-of-cloud aerosol in the size rage below the activation diameter of the cloud.

It would be very valuable to perform a size-resolved analysis of the particles. However, the mass concentrations below approximately 100 nm is so low that the "PToF-mode" of the AMS (that allows for size-resolved analysis) is not able to provide data above the noise level. It has to be mentioned here that the duty cycle of the PToF-mode is by a factor of 50 lower than the normal "MS mode".

a. Page 7, Line 35: The authors state that the mass concentration of the interstitial aerosol is lower than that of the CDR. This would be expected since CDR particles are larger in diameter (hence more mass) than the interstitial aerosol.

Yes, this is to be expected, but it is necessary to mention it here, because for black carbon it is not the case, which surprised us because impactor data at the upwind site showed a large BC fraction in a size range above 400 nm (aerodynamic). But apparently these large BC particles were not activated (here composition appears to matter more than size!) and therefore show up in the interstitial aerosol. Unfortunately, no size-resolved BC data are available from the summit site.

b. Figure 4. It would be useful to see the significance of the difference between the interstitial and CDR composition, through comparison of the similar size fractions (< 90 nm (INT) and > 90 nm (CDR)) during NCE.

As explained above, it is not possible to evaluate the size resolved data for diameters smaller than around 100 nm.

3) Another concern is that the transmission efficiency of the aerodynamic lens used for Aerodyne products sample aerosol particles with "good" efficiency between _90 nm and _300 nm (Liu et al., 2007), however below (and above) these limits the transmission efficiency of the instrument decreases rapidly. One needs to take this transmission efficiency into account and also the implications that this may have on the quality of the AMS data at these lower diameters. Baseline errors will likely have an impact at these diameters so it is necessary to take precautions to ensure that measured aerosol compositions and concentrations at these small diameters are real and not just arbitrary noise.

Peter S. K. Liu, Rensheng Deng, Kenneth A. Smith, Leah R. Williams, John T. Jayne, Manjula R. Canagaratna, Kori Moore, Timothy B. Onasch, Douglas R. Worsnop, and Terry Deshler Transmission Efficiency of an Aerodynamic Focusing Lens System: Comparison of Model Calculations and Laboratory Measurements for the Aerodyne Aerosol Mass Spectrometer Aerosol Science And Technology Vol. 41 , Iss. 8,2007

We fully agree. This is one of the reasons why a size-resolved analysis of the AMS data is not possible for small diameters. Therefore we can only report size-integraded data from the "MS mode". However, the observed differences between interstitial, cloud residual, and out-of-cloud particle composition can not be explained by the lens transmission limitations, thus the findings and conclusions of our paper are not affected.

4) A large part of this manuscript is focused on the enrichment of nitrate in aerosol particles after cloud processing. However, these conclusions were made through comparing CDR with NCE before, and in some cases, after the cloud event. If cloud processing was indeed used

to result in the enrichment of nitrate in particles, would one not expect to observe higher nitrate in aerosol particles once the cloud event has passed?

> We do not expect that this is the case, because the effect is most likely reversible. We detailed our arguments in two sections of the manuscript: In section 3.2.1 we argued:
>> "…while in ambient air, dependent of the gas-phase concentrations of $HNO_3$ and NH3, the situation will be different and a larger part of the nitrate and ammonium may be released back into the gas phase. If the air after the cloud returns to the same temperature and relative humidity conditions as before the cloud, is it to expect that the overall equilibrium between particle phase $NH_4NO_3$ and gas phase $NH_3$ and $HNO_3$ will be equal 30 to that before the cloud, as long as no chemical processing of nitrate and ammonium occurs in the cloud phase."
>
> In the summary we repeated this argument:
>> "In general, cloud processing will tend to evenly distribute nitrate and ammonium over the processed aerosol particles: At the same temperature and relative humidity after the cloud passage as before, it is to be assumed that the same equilibrium between particle-phase ammonium nitrate and gas phase nitric acid and ammonia as before the cloud is established. Thus, the absolute amount of particle phase ammonium nitrate should be the same after the cloud as before the cloud."

5) A constant correction efficiency (CE) of 0.5 was applied to all data. However, there are several periods (shown in Figure 2 a)) where the contribution of nitrate aerosol particles was greater than 25% to the total aerosol mass. In general, within the aerosol mass spectrometry community, it is recommended to apply a composition dependent CE as outlined in the manuscript Middlebrook et al., (2012). Middlebrook,A.N R. Bahreini, J. L. Jimenez, and M. R. Canagaratna (2012) Aerosol Sci. Tech, 46:258– 271.

> We have chosen to apply a constant collection efficiency of 0.5 for simplicity. We are of course familiar with the Middlebrook parameterization. But it must be said that the parameterizations by Middlebrook at al. are based on data sets with a combined uncertainty ($2\sigma$) of 45 %. There are a few time periods in our data set where the composition dependent CE calculated using the Middlebrook formula is significantly higher than 0.5, but it did not change a lot in the overall correlation between the mass inferred from SMPS and the sum of AMS and MAAP when it was tested.
> Furthermore, many parameters used in our analysis are not affected by the CE value: Mass fraction (Fig 4), $f_{44}$, $f_{43}$ (Fig 8, 9, 10), $NO^+/NO_2^+$ (Fig 11).

6) Cloud events listed in table 1 varied from 3 hrs up to 12 hrs. Air mass trajectories were used to verify that there was no change in air mass properties, however could there be more robust criteria used to classify these cloud events. Could the authors incorporate the FSSP cloud droplet distribution and LWC measurements to evaluate whether the cloud properties changed outside a certain limit. For example Fig. 12 shows the cloud droplet diameters and concentrations changing during the cloud event, this was accompanied also by a change in the LWC. How can the authors ensure that these changes in cloud properties were not accompanied by slight air mass changes, or entrainment of new aerosol types. This might influence the comparison with NCE. a. Likewise, how long a time period should be compared from the NCE data? It might not be judicious to include data from 24 hours prior to the measurements.

> Cloud events were selected and defined based on robust criteria, as described in Tilgner et al. (2014). Coefficient of divergence (COD) analyses were performed using continuous measurements of ozone and particle concentration (49 nm diameter size bin). We do not want

to modify these cloud event times, in order to maintain comparability among the HCCT cloud studies.

For non-cloud events it was necessary to find times without clouds as close as possible to the respective cloud event. Only in the case of FCE11.2, the corresponding non-cloud event (NCE 0.8) was more than 24 hours apart from the cloud event itself.

Minor comments: The only mention of orographic clouds is in the title of this manuscript. Although, full details of the experimental design is included in Tilgner et al., some discussion of the importance and how these cloud events were verified as orographic should be included.

See also reply to reviewer #2:
A description of the cloud types is given in detail in other publications in the special issue on the HCCT campaign, e.g., Tilgner et al. (2014) or Roth et al. (2016). Therefore, we added only a brief description of the cloud to section 3.1.:

The whole time series of the cloud droplet number distribution measured by the FSSP is given in Figure S4. The 14 FCE are also indicated by the grey bars. These FCE were chosen based on certain criteria as detailed in Tilgner et al. (2014), focusing on connected flow conditions between the upwind, the summit and the downwind station. These conditions went along with stable south-west flow conditions, thus the clouds were mainly of orographic nature, however in certain cases the meteorological analysis revealed that the clouds were not purely orographic (FCE1.1, FCE2.1, FCE26.1). For details see Table 5 in Tilgner et al. (2014) and Table 1 in Roth et al. (2016).

Page 4, section "Analysis instruments" How was the aerosol dried prior to sampling in the interstitial aerosol?

The interstitial aerosol (and therefore also the out-of-cloud aerosol) was dried using a Nafion dryer. Humidity measured in the sampling line was below 35%. This was added so section 2.2 (Aerosol and cloud sampling at the summit site):

The air sampled by the interstitial inlet was dried using a Nafion dryer. Measured relative humidity in the sampling line behind the dryer was below 35%.

Page 13, Line 33: The authors mention that biogenic emissions could be a source of the higher OA measured at higher temperatures. Is their any evidence of biogenic emissions during these periods? Gas-phase measurements, lower than average BC concentrations, etc.

No, there is no evidence for biogenic emissions, only indications: The field site is surrounded by forest, and in the upwind direction there are no large cities. In the supplement to Roth et al. (2016) (http://www.atmos-chem-phys.net/16/505/2016/acp-16-505-2016-supplement.pdf) there is a map and a list of cities within a 50 km radius around the field site. Thus, the assumption of biogenic emissions is plausible but can't be proven.

Page 16, Line 29: What is the significance of these correlations? 99%, 95% and how is the 'significance' determined?

Significance was determined using both t-test and F-test (using Wavemetric's IGOR function "statslinearregression") with 95% confidence interval. All correlations but Graph 6 ($NO_3$ vs LWC in $2^{nd}$ half of cloud) are significant.

References

Dusek, U., Frank, G. P., Hildebrandt, L., Curtius, J., Schneider, J., Walter, S., Chand, D., Drewnick, F., Hings, S., Jung, D., Borrmann, S., and Andreae, M. O.: Size matters more than chemistry for cloud-nucleating ability of aerosol particles, Science, 312, 1375-1378, 2006.

Henning, S., Dieckmann, K., Ignatius, K., Schäfer, M., Zedler, P., Harris, E., Sinha, B., van Pinxteren, D., Mertes, S., Birmili, W., Merkel, M., Wu, Z., Wiedensohler, A., Wex, H., Herrmann, H., and Stratmann, F.: Influence of cloud processing on CCN activation behaviour in the Thuringian Forest, Germany during HCCT-2010, Atmos. Chem. Phys., 14, 7859-7868, doi: 10.5194/acp-14-7859-2014, 2014.

Roth, A., Schneider, J., Klimach, T., Mertes, S., van Pinxteren, D., Herrmann, H., and Borrmann, S.: Aerosol properties, source identification, and cloud processing in orographic clouds measured by single particle mass spectrometry on a central European mountain site during HCCT-2010, Atmos. Chem. Phys., 16, 505-524, doi: 10.5194/acp-16-505-2016, 2016.

Tilgner, A., Schöne, L., Bräuer, P., van Pinxteren, D., Hoffmann, E., Spindler, G., Styler, S. A., Mertes, S., Birmili, W., Otto, R., Merkel, M., Weinhold, K., Wiedensohler, A., Deneke, H., Schrödner, R., Wolke, R., Schneider, J., Haunold, W., Engel, A., Wéber, A., and Herrmann, H.: Comprehensive assessment of meteorological conditions and airflow connectivity during HCCT-2010, Atmos. Chem. Phys., 14, 9105-9128, doi: 10.5194/acp-14-9105-2014, 2014.

Wu, Z. J., Poulain, L., Henning, S., Dieckmann, K., Birmili, W., Merkel, M., van Pinxteren, D., Spindler, G., Müller, K., Stratmann, F., Herrmann, H., and Wiedensohler, A.: Relating particle hygroscopicity and CCN activity to chemical composition during the HCCT-2010 field campaign, Atmos. Chem. Phys., 13, 7983-7996, doi: 10.5194/acp-13-7983-2013, 2013.